# Modelling the sequential earthquake-tsunami response of coastal road embankment infrastructure

Azucena Román-de la Sancha[1], Rodolfo Silva[2], Omar S. Areu-Rangel[1], Manuel Gerardo Verduzco-Zapata[3], Edgar Mendoza[2], Norma Patricia López-Acosta[2], Alexandra Ossa[2], Silvia García[2]

1 Graduate student, Instituto de Ingeniería, Universidad Nacional Autónoma de México, Circuito Escolar, Ciudad Universitaria, 04510, Coyoacán, Cd. Mx., México

2 Researcher, Instituto de Ingeniería, Universidad Nacional Autónoma de México, Circuito Escolar, Ciudad Universitaria, 04510, Coyoacán, Cd. Mx., México

3 Researcher, Facultad de Ciencias Marinas, Universidad de Colima, Carretera Manzanillo-Cihuatlán Km 19.5, Colonia El Naranjo. 28868 Manzanillo, Colima, México

*Correspondence to*: Azucena Román-de la Sancha (azucena.roman@ingenieria.unam.edu)

**Abstract.** Transport networks in coastal, urban areas are extremely vulnerable to seismic events, with damage likely due to both ground motions and tsunami loading. Most existing models analyse the performance of structures under either earthquakes or tsunamis, as isolated events. This paper presents a numerical approach that captures the sequential earthquake-tsunami effects on transport infrastructure in a coastal area, taking into consideration the combined strains of the two events. Firstly, the dynamic cyclical loading is modelled, applied to the soil-structure system using a finite difference approximation to determine the differential settlement, lateral displacement, and liquefaction potential of the foundation. Next, using a finite volume method approach, tsunami wave propagation and flooding potential are modelled. Finally, the hydrostatic and hydrodynamic loads corresponding to the wave elevation are applied to the post-earthquake state of the structure, to obtain a second state of deformation. The sequential model is applied to an embankment in Manzanillo, Mexico, which is part of a main urban road, the response is analysed using ground motion records of the 1995 Manzanillo earthquake-tsunami event.

## 1.      Introduction

Transportation networks are key elements of the economy and society; therefore, more attention has been paid to the need for sustainable, adaptive and resilient transport systems under natural hazards and other emergency conditions in the last decades. Past experience of the subduction mechanisms of seismic events has shown the vulnerability of transport infrastructure located in coastal urban areas associated with the cascade effect of earthquake-tsunami-flooding-ground motion (Goda 2021; Goda et al. 2021; Williams et al., 2020, Williams et al., 2019; Sarkis et al., 2018; Bhattacharya et al., 2017; Rowell and Goodchild, 2017; Goda et al. 2017; Koshimura et al., 2014). Existing methodologies to analyse the response of transport assets during earthquake and tsunami conditions consider these phenomena as independent events. Probabilistic analyses and fragility curves (Akiyama et al., 2020; Burns Patrick et al., 2021) are examples of multi-hazard approaches that aim to determine the occurrence probability of both phenomena simultaneously and the expected level of damage of structures after them. Recent multi-hazard literature has focused on structures in the urban environment such as buildings and bridges (Attary et al., 2021; Burns Patrick et al., 2021; Ishibashi et al., 2021; Karafagka et al.,

2018), while little attention has been paid to the response of road networks to sequential earthquake-tsunami processes (Iai, 2019).

The impacts of a combined earthquake-tsunami on urban transport networks have important economic and social effects in the short, medium and long terms, in both human and material losses. These impacts are fundamentally associated with the loss of connectivity of regions or cities, which primarily disturb the

continuity of rescue efforts, reconstruction, urban logistics, health and safety activities (Dizhur et al., 2019; Sarkis et al., 2018). Most of the negative effects are associated with partial or total damage to urban and interurban roads that disrupt land communication between key sites (Cubrinovski, 2013). In coastal areas, one significant factor causing damage in earthquakes is the increase in the pressure within the soil mass induced by the seismic loads, and by the accumulation of hydrodynamic pressure in the non-cohesive saturated soils

common in these areas. From observations of past events in coastal areas it is seen that moderate seismic intensity can cause liquefaction, resulting in reduced stiffness and a loss of shear strength in liquefied soils. This can lead to soil settlement, increased lateral ground pressure, and loss of strength (Kakderi and Pitilakis, 2014), bringing about damage to structures, sometimes even causing their collapse, and in the case of embankments, potential slope-stability problems, or even landslides.

Regarding tsunami damage, one of the most critical aspects to consider is the hydrodynamic behaviour of the wave as it approaches the coast. However, information on the physical parameters that characterize these waves is often limited, due to difficulties in achieving accurate measurements during the event.

The impact of a tsunami on the coast is governed by non-linear physics, such as turbulence (Klapp et al., 2020). The hydrodynamic behaviour during a tsunami is complex and can significantly affect structures. Forces

induced by the hydrodynamics associated with a tsunami are governed by various fluid parameters, such as density, velocity, and depth, as well as the geometry of the structure. These forces induced by the pressures and velocities of tsunamis are particularly important in the stability of coastal structures, affecting assets such as piers, embankments and bridge piles (Chinnarasri et al., 2013). Numerical approaches based on finite element methods (Argyroudis and Kaynia, 2015; Mckenna et al., 2021) and finite differences (Mayoral et al., 2016;

Mayoral et al., 2017) have been applied to assess the seismic vulnerability and potential damage of transport infrastructure. Likewise, the effects of hydrodynamic loads on structures have been estimated through modelling methods, such as finite volume method (Jose et al., 2017) and smooth particle hydrodynamic method (Altomare et al., 2015; Klapp et al., 2020, Hasanpour et al. 2021).

Besides the simulation models available to assess the individual effects of earthquake and tsunami loads on

transport infrastructure, a few, limited, modelling approaches to predict sequential damage and effects are available. From a review of the literature on fragility and vulnerability models it was seen that most studies focus on individual transport components or networks, usually considering only one hazard at a time (Argyroudis and Kaynia, 2015; Argyroudis and Mitoulis, 2021; Briaud and Maddah, 2016; Iai, 2019; Maruyama et al., 2010; Nibs, 2004). The studies mentioned focus mainly on the vulnerability of bridges and tunnels, and

the main emphasis is on ground movement due to seismic excitation. Research based on simplified models has examined other structures such as embankments, slopes, retaining walls, and abutments. Existing models for

these elements are based on empirical data or expert judgment, mainly for seismic analysis, and focus on a universe of limited typologies.

There is, at present, the opportunity to improve the modelling approaches that consider more structure types, as well as multiple hazards, including earthquakes and tsunamis simultaneously. To extend these studies, the risk analysis can be extended by using one of the methodologies described by Escudero Castillo et al. (2012) or better yet, by implementing an analytical framework similar to the Drivers-Exchanges-State of the environment-Consequences-Responses (DESCR) framework proposed by (Escudero Castillo et al., 2012; Silva et al., 2020) in which ecosystem-based adaptation and community-based adaptation is considered (Silva et al., 2019).

This paper proposes a numerical approach to capture the sequential earthquake-tsunami effects on urban road embankments located in coastal areas. The approach allows the effective stress paths and corresponding soil strain evolution during the event to be considered, taking into account soil-embankment response, tsunami propagation, and hydrodynamic behaviour. The first step consists of computing the initial static stress and deformation conditions, and modelling the dynamic cyclic loading applied to the soil-embankment system in

an earthquake. A finite difference approximation is used to determine the differential settlements and lateral displacements, associated with excess pore pressure generation during cyclic loading. This allows the assessment of the liquefaction potential of the soil foundation. Next, a finite volume approach and smooth particle hydrodynamics are combined, to model tsunami wave propagation at oceanic level and flooding potential. Finally, the resulting sea-surface elevation and hydrodynamic loads are applied to the post-earthquake

state of the structure to obtain the final deformation state.

## 2.    Sequential earthquake-tsunami response model on transport infrastructure

The sequential model proposed is comprised of three steps, Figure 1. The first step is the numerical analysis of the seismic response of the soil-structure system. The next stage corresponds to the wave generation and propagation model from deep, offshore waters up to the coast. Finally, there is the analysis of the soil-structure

system response to hydrodynamic loadings, given the state of deformation induced by the seismic excitation. This methodology is applied to an embankment of urban road, in the coastal city of Manzanillo, Mexico.

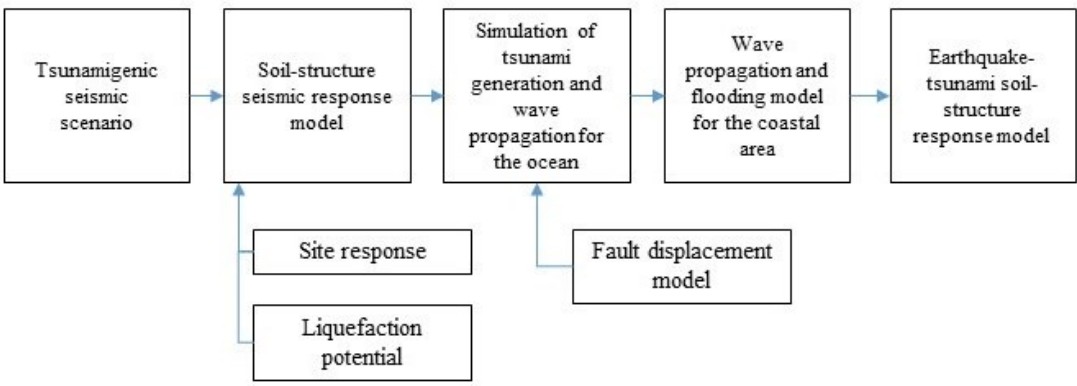

**Figure 1:  Evaluation of the sequential earthquake-tsunami effects on transportation infrastructure.**

### 3.    Case study

Manzanillo, on the Mexican Pacific Coast, is the most important commercial port in the country. The zone is seismically complex, due to the triple junction of the Cocos and Rivera subduction plates that are moving towards and below the continent, on the North American Plate, Figure 2. The city has been affected by many major earthquakes, and several in the last century (Tonatiuh Dominguez et al., 2017; Eqe International, 1996; Ovando-Shelley and Romo, 2004). In 1995, an earthquake Mw = 8 caused significant damage, mainly in the

coastal area of the city, where collapsed buildings were reported and liquefaction of the saturated sandy soils was observed at several locations in the port area of Manzanillo.

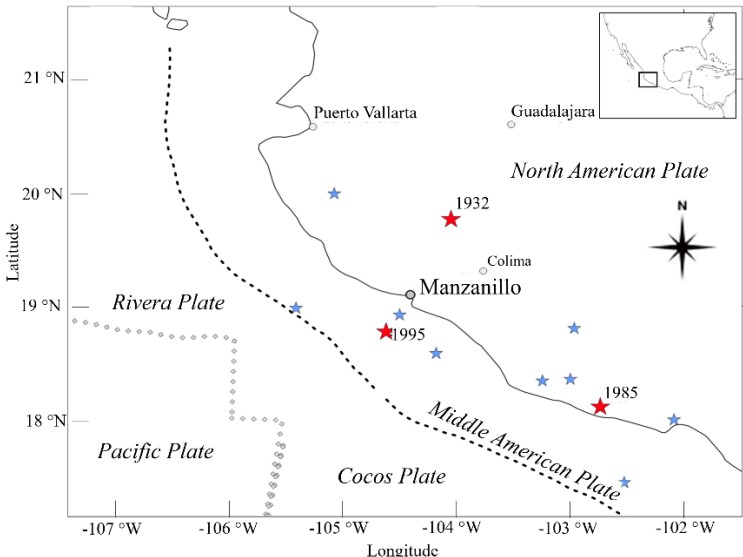

**Figure 2: Seismic environment of the study area. Red stars indicate earthquakes M ≥ 8.0, blue stars indicate earthquakes 7.0 ≤ M ≤ 8.0.**

The urban area of Manzanillo and its commercial port are connected to the rest of Mexico via the Colima-Manzanillo, Manzanillo-Puerto Vallarta and Cuyutlán-Manzanillo highways. Part of the embankment on one of these main avenues was selected for the analyses performed in this research. The Boulevard Miguel de la Madrid, runs parallel to Miramar Beach, which is on Santiago Bay, NW of the city of Manzanillo, Figure 3. This beach has attractive natural characteristics, perfect for practicing sun and beach recreation activities

(Cervantes et al., 2015); the road has become a waterfront promenade, with several restaurants and access points to the beach along the road. This road also connects the communities of El Naranjo, in the west, and Colinas de Santiago, in the east. It is also an important public transport route, serving tourists and the labour force going to and from the restaurants, hotels and recreation facilities. Further west is Manzanillo International Airport. This road is therefore a vital part of the infrastructure of the city, facilitating many social-economic activities.

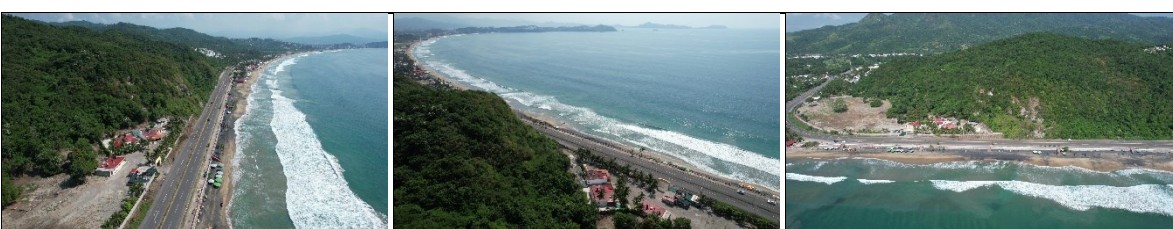

**Figure 3: Aerial views of Boulevard Miguel de la Madrid.**

The data on the topography and topology of the section of embankment analysed here were obtained in field work carried out in 2021. A total of 24 topographic profiles along a 2 km long section were surveyed, Figure 4. The surveying equipment eSurvey E300 PRO + P8II GNSS was positioned at a control point to later link the collected data with the official datum. With a centimetre precision, post-processing of the data gave the coordinates of a geodesic point which served as a reference to carry out the topographic survey. Based on the geometric characteristics of these cross-sections and their relative importance, the cross-section shown in Figure 5 was selected for the sequential earthquake-tsunami analysis. The soil-embankment system was modelled using a three-dimensional approach where the 20 m segment of the coastal profile selected was considered uniform. This was the section considered to have the most critical condition of the road due to the height and the slope of the embankment.

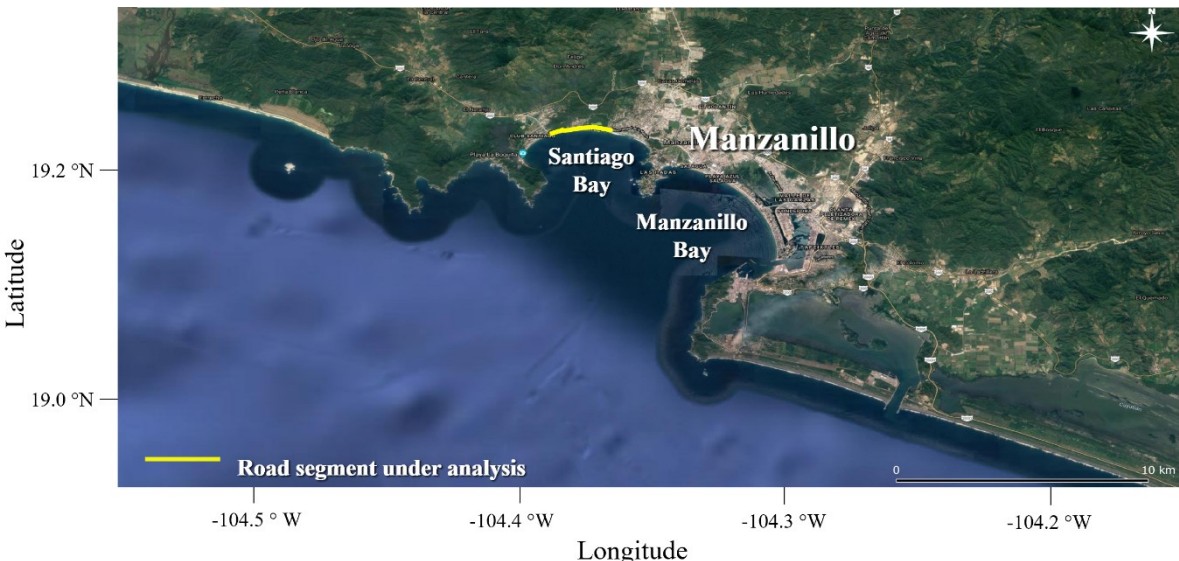

**Figure 4: Location of the road embankment analysed (© Google Earth).**

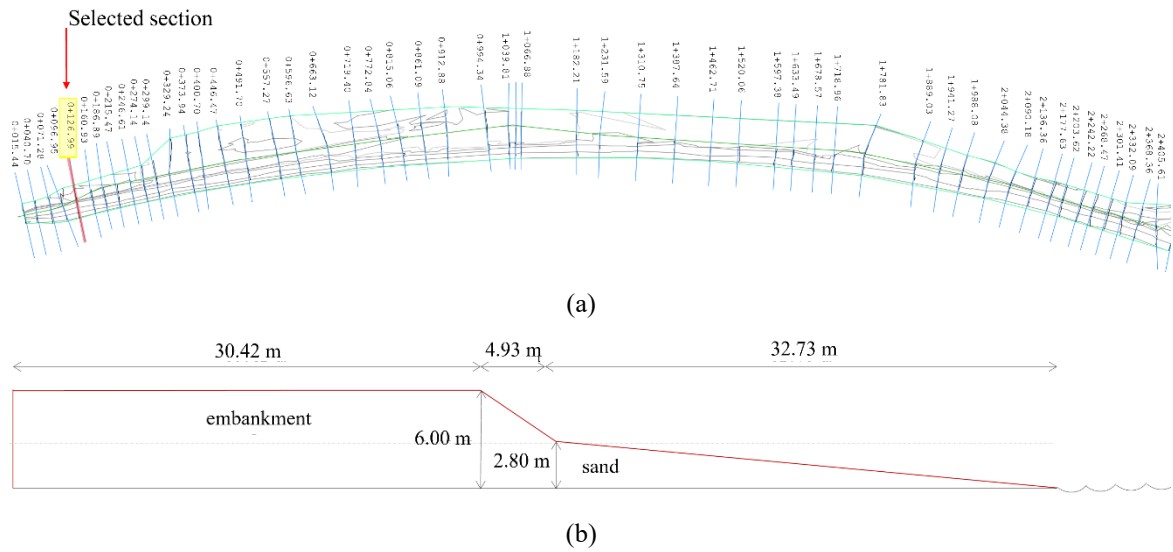

**Figure 5: (a) Cross sections taken along the road and (b) the cross-section used in the analysis.**

### 3.1.    Seismic and geotechnical characterization of the study area

Manzanillo is in the northern part of the Sierra Madre del Sur mountain range. Geological studies by Tonatiuh Dominguez et al. (2017) state that granite intrusions, extrusive igneous outcrops and limestone deposits are found in the region. Deposits of clay and organic materials are also found in the two saltwater lagoons nearby. Sand deposits are found along the coast and alluvial deposits are found around the igneous deposits, Figure 6. Based on geophysical explorations at points S1 to S5, Figure 6, (Tonatiuh Dominguez et al., 2017) shows an in situ characterization of some dynamic properties, such as shear wave velocities and the shear wave velocities obtained from exploration at Site 4, which is considered representative for characterizing sites with soil type B, where the section of the road embankment studied is located.

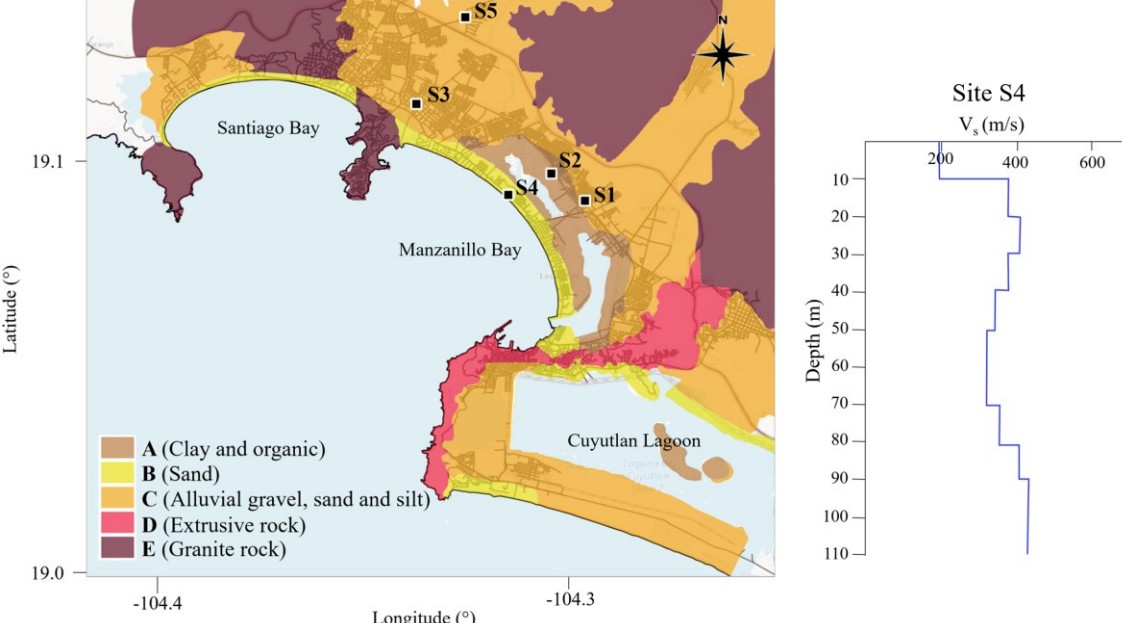

**Figure 6: Survey sites S1 - S5 and geotechnical characterization of Manzanillo and Santiago Bays obtained from (Tonatiuh Dominguez et al., 2017) and Shear wave velocity at Site S4**

Manzanillo is located in the subduction zone between the Rivera and Cocos tectonic plates with the North American plate. The rate of displacement at the contact surfaces is about 50 to 70 mm per year, (Demets and Wilson, 1997), so the probability of severe earthquakes is high. Furthermore, in subduction zones, there is a high likelihood of tsunami occurrence, given the vertical displacement generated on the seabed, which can be several meters, extending over tens of thousands of square kilometres. Over time, numerous serious earthquakes have been recorded in this area, causing high intensity ground movements, both in the continental region and along the coastline, as well as tsunami damage on the coast. According to historical records, at specific sites in this area the wave heights associated with tsunamis have reached 10 m. Tsunamigenic events occurring in this region are of Mw>8 magnitudes, among which are the 1787, 1932, 1985 and 1995 earthquakes in the study area.

### 3.2. Tsunamigenic seismic scenario

The seismic features were characterized using records from the seismic station MZ01, Figure 7, located on sandy deposits. It is the only station in free-field conditions in Manzanillo that was operating during the 1995 event (Mw = 8.0). Table 1 summarizes the characteristics of the event. Figure 7 shows the location of station MZ01 and the epicenter of the 1995 event. Figures 8 and 9 show the acceleration time histories and their corresponding response spectra for each parameter measured at station MZ01.

**Table 1 Characteristics of the 1995 Manzanillo earthquake and the MZ01 station record.**

| Site | Event | Magnitude | Epicentral distance (km) | Soil | *PGA* (gal) |
|------|-------|-----------|--------------------------|------|-------------|
| COLIMA (MZ01) | 09/10/1995 | $M_w$=8.0 | 47.5 | Sand | N00E=380.25 V=187.58 N90E=402.9 |

Note: *PGA*: Peak ground acceleration

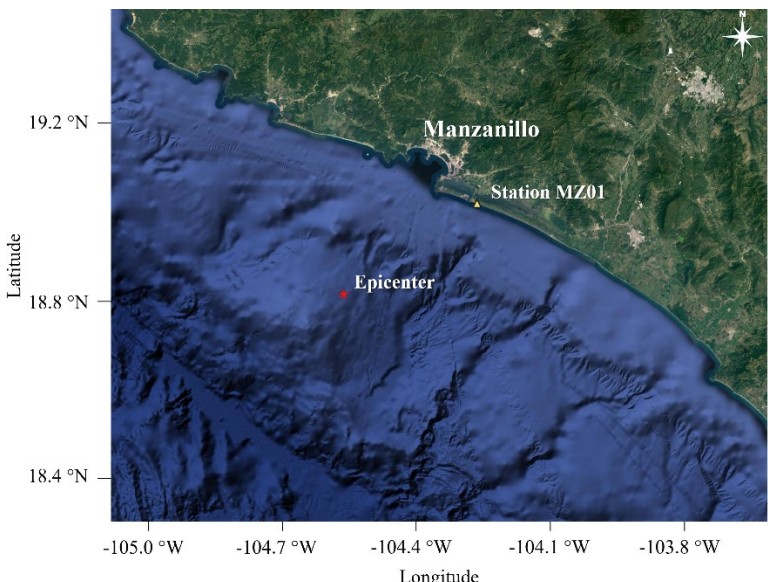

**Figure 7: Location of the seismic station MZ01 and the epicentre of the 1995 Manzanillo earthquake (Mw=8) event (© Google Earth).**

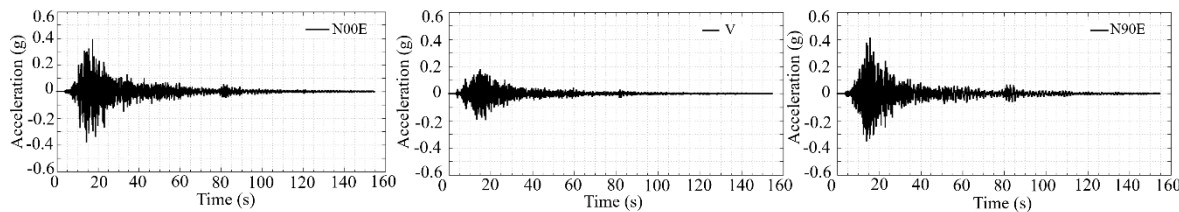

**Figure 8: Acceleration time histories at station MZ01 during the 1995 Manzanillo earthquake (Mw=8.0) earthquake.**

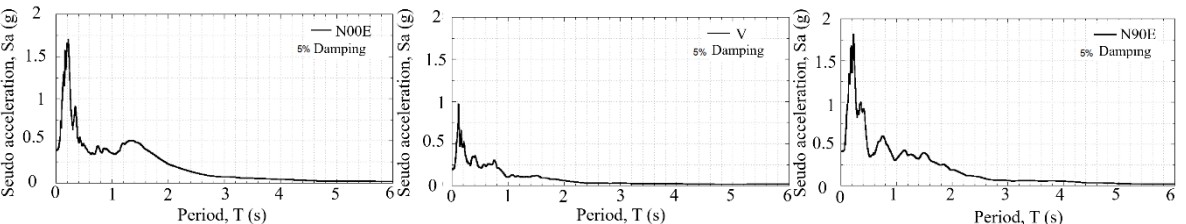

**Figure 9: Response spectra from the time histories recorded at station MZ01 during the 1995 Manzanillo earthquake (Mw=8.0) earthquake.**

To perform the analysis, the record of N90E component showing the larger accelerations and pseudo-accelerations was selected. The accelerations time history was baseline corrected to avoid residual velocities or displacements at the end of the movement, which could be confused with permanent displacements. Figures 10 (a) and (b) show the acceleration, velocity, and displacement histories before and after correction, respectively. To simulate wave propagation during the earthquake, the selected acceleration history (N90E) was deconvolved using the soil profile of site S4 (Figure 6), using the code SHAKE (Schnabel et al., 1972), which determines the vertical propagation of shear waves in a horizontally stratified semi-infinite soil deposit. Subsequently, the deconvolved earthquake was propagated again upward to verify that the propagation of the deconvolved and the recorded history are consistent. Figure 11 presents the comparison between the recorded and calculated response spectrum.

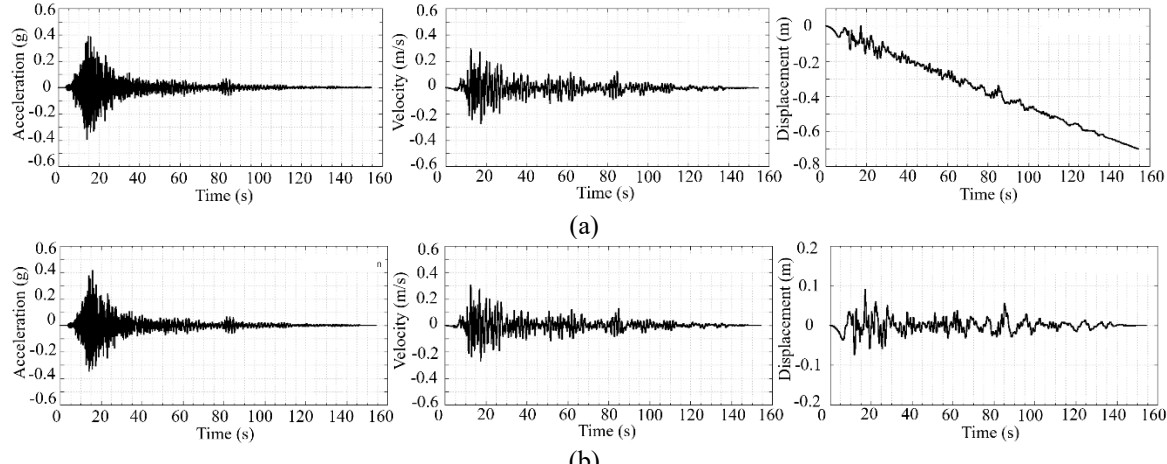

**Figure 10: Acceleration, velocity and displacement time histories (a) without base line correction (b) with base line correction.**

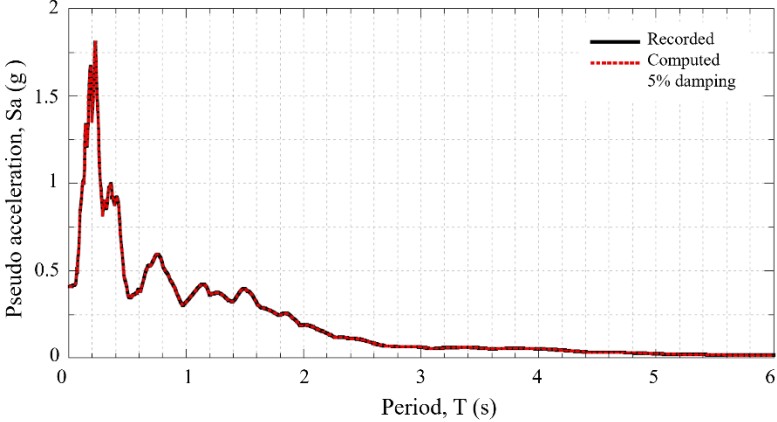

**Figure 11: Comparison between the recorded and calculated spectrum after the propagation of the deconvolved earthquake.**

### 3.3. Step 1: seismic response

To simulate seismic wave propagation, during the earthquake, a site response analysis was performed using the N90E acceleration history. The record was deconvolved and propagated through the soil profile of the S4 site, performing a one-dimensional deterministic analysis. In this way, a first approximation of the expected dynamic amplification in the soil deposit was obtained, which is useful prior to the non-linear analysis in the time domain. This analysis also allows the maximum dynamic shear stresses in the soil to be obtained, which are used to estimate the liquefaction potential in the simplified methods. Since dynamic laboratory tests were not performed on samples obtained near the study area, it was considered appropriate to assume the damping curves and shear stiffness modulus ratio presented by Seed and Idriss (1970) for sands, Figure 12.

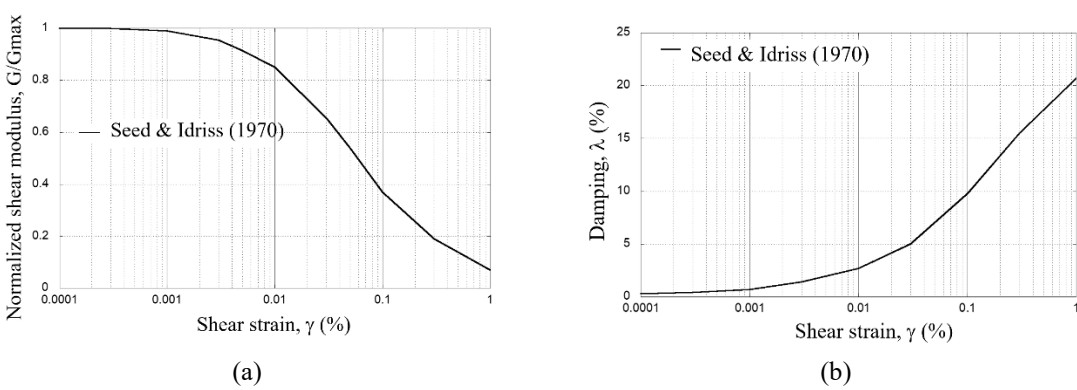

(a)                                                                 (b)

**Figure 12: (a) Normalized shear stiffness curves $G/G_{max}$ and (b) damping curves $\lambda$**

The stress history was determined and transformed into an equivalent number of cycles of uniform shear stress. In this way, the intensity and duration of ground motions are considered, as well as the variations of shear stress with depth. For practical purposes, the equivalent number of cycles of shear stress, $\tau_{av}$ , can be estimated as 65% of the maximum stress, $\tau_{max}$ (Seed and Idriss, 1970), Figure 13.

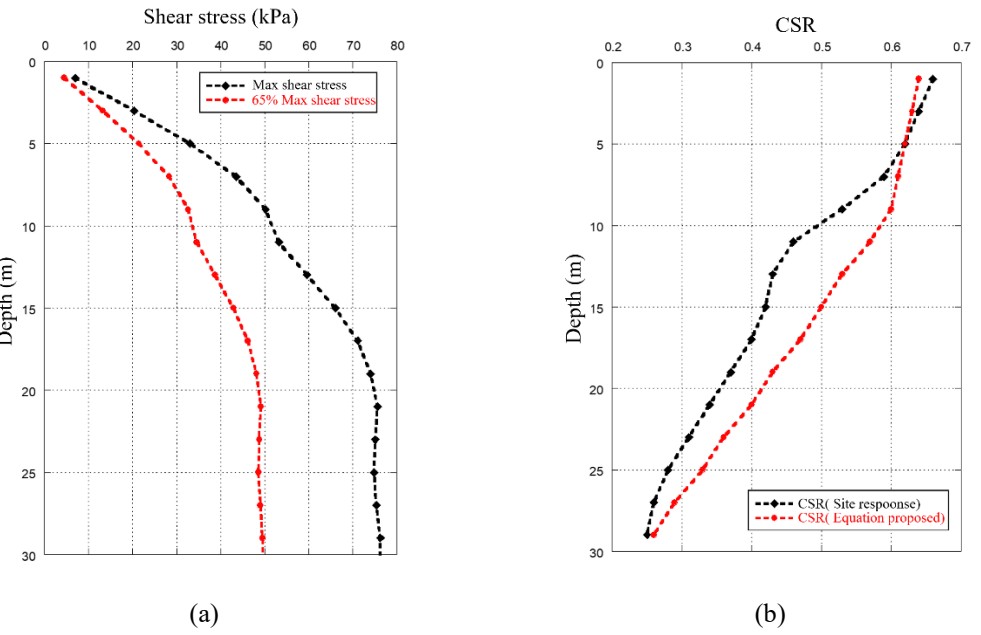

(a)                                                                 (b)

**Figure 13: (a) Maximum shear stress profile for the study site, (b) cyclic stress ratio profile, using the equation proposed by Seed and Idriss (1971) and the site response analysis.**

The number of equivalent stress cycles *Neq* depends on the magnitude and duration of the earthquake. Seed et al. (1975) applied a weighting procedure to establish a uniform number of cycles of shear stress *Neq* (with an amplitude of 65% of the maximum cyclic shear stress $\tau cyc = 0.65\tau max$) that would produce an equivalent pore pressure, where the number of cycles of uniform shear stress increases with increasing magnitude of the earthquake. For the case study, an earthquake magnitude Mw = 8.0 corresponds to an equivalent number of

cycles of uniform shear stress equal to 21, Figure 14.

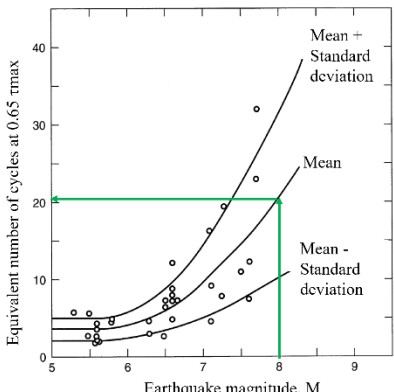

**Figure 14: Equivalent number of uniform cyclic loading *Neq* for different seismic magnitudes (Martin et al. 1975)**

The liquefaction potential of the soil was evaluated using the cyclic stress criterion (Kramer, 1996; Seed and

Idriss, 1970). One of the most common methods to evaluate liquefaction resistance is based on standard penetration tests (SPT). This criterion uses the cyclical resistance ratio CRR from blow counts from the SPT test, corrected to 60% of the energy N60. Thus, according to the shear wave velocity profile of the study site, $N_{60}$ was estimated using the expressions and parameters shown in Tables 2 and 3, obtaining $N_{60}$=27, for the first 10 m of depth.

**Table 2 Correlations to estimate shear wave velocity.**

| Reference | Correlation | $N_{60}$ |
|---|---|---|
| (Pitilakis et al., 1999) | $V_s = 145N_{60}^{0.178}$ | $N_{60} = (V_s/145)^{(1/0.178)}$ |
| (Dikmen, 2009) | $V_s = 73N^{0.33}$ | $N = (V_s/73)^{(1/0.33)}$ |
| (Imai, 1977) | $V_s = 80.6N^{0.331}$ | $N = (V_s/80.6)^{(1/0.331)}$ |

Note: *N*: SPT blow counts, $N_{60}$: blow counts 60% energy corrected.

**Table 3 Parameters for estimating N60.**

| Depth | | (Pitilakis et al., 1999) | (Dikmen, 2009) | | (Imai, 1977) | |
|---|---|---|---|---|---|---|
| from (m) | to (m) | $N_{60}$ | N | $N_{60}$ | N | $N_{60}$ |
| 0 | 10 | 25 | 45 | 32 | 33 | 24 |
| 10 | 20 | 214 | 145 | 103 | 106 | 75 |
| 20 | 30 | 65 | 76 | 54 | 56 | 40 |

The safety factor against liquefaction for the embankment $SF_l$, was calculated based on the critical strength ratio CSR, which was derived from the site response analysis. A correction factor for the magnitude of the earthquake was applied. The CRR was obtained using the Robertson and Wride (1998) approximation, and the magnitude

correction factor, MSF, proposed by Andrus and Stokoe (1999): was considered through the following equations

$$CRR_{7.5} = \frac{1}{34-(N_1)_{60}} + \frac{(N_1)_{60}}{135} + \frac{50}{[10*(N_1)_{60}+45]^2} - \frac{1}{200} \tag{1}$$

$$MSF = \left(M_w/7.5\right)^{-2.56} \tag{2}$$

$$SF_l = \left(CRR_{7.5}/CSR\right)MSF \tag{3}$$

The preliminary result shows that in the first 10 m of the profile there is the potential for soil liquefaction.

**Table 4 Safety factor against liquefaction**

| Average depth (m) | CSR | CRR$_{7.5}$ | MSF | SF$_l$ |
|---|---|---|---|---|
| 1 | 0.66 | 0.34 | 0.85 | 0.44 |
| 3 | 0.64 | 0.34 | 0.85 | 0.45 |
| 5 | 0.62 | 0.34 | 0.85 | 0.46 |
| 7 | 0.59 | 0.34 | 0.85 | 0.49 |
| 9 | 0.53 | 0.34 | 0.85 | 0.54 |
| 11 | 0.46 | Not liquefiable | 0.85 | >1.0 |
| 13 | 0.43 | Not liquefiable | 0.85 | >1.0 |
| 15 | 0.42 | Not liquefiable | 0.85 | >1.0 |
| 17 | 0.40 | Not liquefiable | 0.85 | >1.0 |
| 19 | 0.37 | Not liquefiable | 0.85 | >1.0 |
| 21 | 0.34 | Not liquefiable | 0.85 | >1.0 |
| 23 | 0.31 | Not liquefiable | 0.85 | >1.0 |
| 25 | 0.28 | Not liquefiable | 0.85 | >1.0 |
| 27 | 0.26 | Not liquefiable | 0.85 | >1.0 |
| 29 | 0.25 | Not liquefiable | 0.85 | >1.0 |

Note: *CSR*: critical strength ratio, *CRR$_{7.5}$*: cyclical resistance ratio, *MSF*: magnitude correction factor, and *SF$_l$*: safety factor against liquefaction.

The seismic response of the embankment was evaluated using a three-dimensional finite difference simulation. This was carried out using the Finn (1970) constitutive model for liquefiable soils, available in the software FLAC$^{3D}$ for the site response analyses of the slope, and to account for soil nonlinearities and loss of strength

associated with pore pressure generation during cyclic loading Figure 15. For monotonic loading, an elastoplastic Mohr-Column failure criterion was used that enables horizontal and vertical accelerations, displacements, shear forces and pore pressure to be obtained. The model was composed of 13,632 elements and 16,083 nodes. The dimension of the element was selected based on the geometry and thicknesses of the soil layers. However, numerical distortion of the propagating wave can occur in a dynamic analysis as a result of

the modelling conditions, (Itasca Consulting Group, 2009). To represent wave transmission accurately, Kuhlemeyer & Lysmer (1973) suggest keeping the size of the spatial element, *Δl*, less than one-fifth of the wavelength associated with the highest frequency component of the input wave that contains significant energy, *fmax* (i.e. *Δl≤λ/5*), where the shortest wavelength *λ* is obtained from *λ=V$_s$/f$_{max}$*. In the case study, the smallest average shear wave velocity *V$_s$* of the site corresponds to that of the first stiff soil layer (i.e. shear wave velocity

is around 260 m/s in the first 10 m layer), while the highest excitation at which the energy is concentrated is about 1-5 Hz. Then, *λ* varies between 260 and 52 m approximately. The maximum spectral responses of the

excitation occur even at higher frequencies (i.e. 0.5 and 10 Hz) as shown in Figure 16. Hence, a $\Delta l$ of 2 m was deemed appropriate. In previous research, using equivalent linear properties (Mayoral et al. 2017; Mayoral et al. 2016; Mayoral et al. 2015), meshes with element sizes of 2 m showed good agreement between finite difference models developed with FLAC$^{3D}$ and SHAKE.

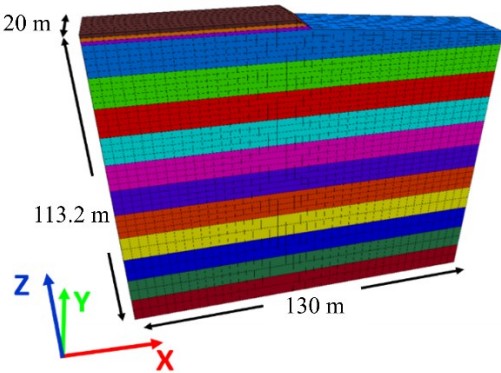

**Figure 15: Soil profile and road embankment in the three-dimensional finite difference model. Each colour represents soil layers according to depths indicated in Table 5.**

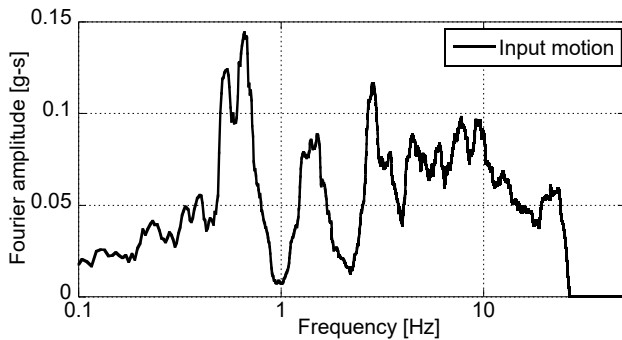

**Figure 16: Frequency content of the input ground motion.**

**Table 5 Soil characteristics.**

| Material | Depth (m) | Density (kg/m³) | Poisson ratio | $C$ (kNm²) | $\varphi$ (°) | $V_s$ (m/s) | $E$ (kN/m²) | $G$ (kN/m²) | $K$ (kN/m²) |
|---|---|---|---|---|---|---|---|---|---|
| 4 | 0-10 | 1.7 | 0.3 | 4.9 | 30 | 257 | 291,937 | 112,283 | 243,280 |
| 5 | 10-20 | 1.7 | 0.3 | 4.9 | 35 | 377 | 628,210 | 241,619 | 523,508 |
| 6 | 20-30 | 1.7 | 0.3 | 4.9 | 35 | 305 | 411,170 | 158,142 | 342,642 |
| 7 | 30-40 | 1.7 | 0.3 | 4.9 | 35 | 332 | 487,190 | 187,381 | 405,992 |
| 8 | 40-50 | 1.7 | 0.3 | 4.9 | 35 | 424 | 794,610 | 305,619 | 662,175 |
| 9 | 50-60 | 1.7 | 0.3 | 4.9 | 35 | 510 | 1,149,642 | 442,170 | 958,035 |
| 10 | 60-70 | 1.7 | 0.3 | 4.9 | 35 | 572 | 1,446,153 | 556,213 | 1,205,128 |
| 11 | 70-80 | 1.7 | 0.3 | 4.9 | 38 | 610 | 1,644,682 | 632,570 | 1,370,568 |
| 12 | 80-90 | 1.7 | 0.3 | 4.9 | 38 | 628 | 1,743,177 | 670,453 | 1,452,648 |
| 13 | 90-100 | 1.7 | 0.3 | 4.9 | 38 | 656 | 1,902085 | 731,571 | 1,585,071 |

| 14 | 100-110 | 1.7 | 0.3 | 4.9 | 38 | 646 | 1,844,537 | 709,437 | 1,537,114 |

Note: $C$: soil shear strength, $\varphi$: friction angle, $V_s$: shear wave velocity, $E$: Young's modulus, $G$: shear stiffness, $K$: constraint modulus.

**Table 6 Embankment characteristics.**

| Material | Height (m) | Density (kg/m³) | Poisson ratio | $C$ (kN/m²) | $\varphi$ (°) | $V_s$ (m/s) | $E$ (kN/m²) | $G$ (kN/m²) | $K$ (kN/m²) |
|---|---|---|---|---|---|---|---|---|---|
| 1 | 0-1.3 | 1.4 | 0.35 | 4.9 | 30 | 285 | 307,968 | 114,062 | 342,187 |
| 2 | 1.3-2.6 | 1.4 | 0.35 | 4.9 | 30 | 264 | 263,334 | 97,531 | 292,593 |
| 3 | 2.6-4.0 | 1.4 | 0.35 | 4.9 | 30 | 241 | 218,700 | 81,000 | 243,000 |

Note: $C$: soil shear strength, $\varphi$: friction angle, $V_s$: shear wave velocity, $E$: Young's modulus, $G$: shear stiffness, K: constraint modulus.

For calibration purposes, one-dimensional models were developed with the program FLAC$^{3D}$, to solve the equation of motion in the time domain, considering both equivalent linear and non-linear properties, and the results were compared with those obtained in the frequency domain, with the program SHAKE (Figure 17). The sig3 model available in FLAC was used to approximately represent soil non-linearity. Thus, the sig3 hysteretic model was used to address the variation of the stiffness modulus and the damping ratio during the seismic event. This model considers an ideal soil, in which the stress depends only on the deformation, and not on the number of cycles loads. With these assumptions an incremental constitutive relationship of the degradation curve can be described by $\tau n / \gamma = G / G_{max}$, where $\tau n$ is the normalized shear stress, $\gamma$ is the shear strain, and $G/G_{max}$ is the normalized secant modulus. The sig3 model is defined using the following expression:

$$\frac{G}{G_{max}} = \frac{a}{1+\exp\left(-\frac{L-x_0}{b}\right)} \tag{4}$$

where $L$ is the logarithmic deformation, defined as $L = log10\ (\gamma)$, and the parameters $a$, $b$ and $x_0$, used by the sig3 model, were obtained through an iterative process, in which the modulus degradation curves were fitted with the model. The corresponding damping is given directly by the hysteresis loop during cyclic loading. For the case study, the parameters a, b and $x_0$ take the values 1.014, −0.50 and −1.25, respectively. Figure 17 shows a comparison between the curves used in the deterministic unidimensional model (Seed and Idriss, 1970), and those obtained with the sig3 model. Figure 18 shows a comparison between the response spectra calculated on the surface with the deterministic one-dimensional model, the equivalent linear FLAC$^{3D}$, and the non-linear FLAC$^{3D}$. Good agreement between the results was observed.

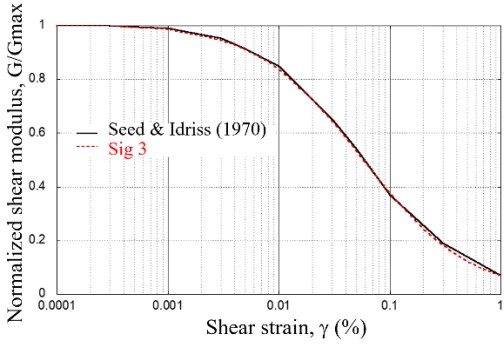
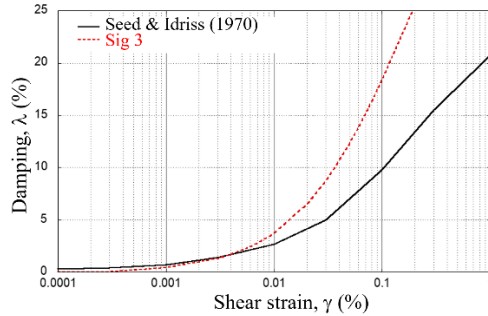

**Figure 17: Results obtained from (Seed and Idriss, 1970) and sig3 model.**

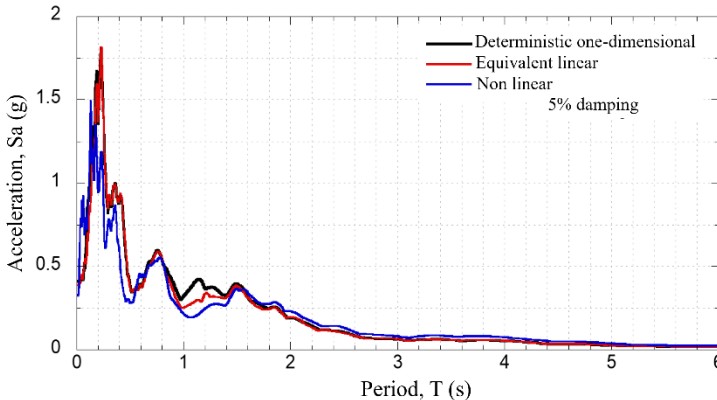

**Figure 18: Response spectra at the surface with the different types of analyses.**

For the three-dimensional model, the Finn-Byrne model was used to evaluate the liquefaction potential. This model is defined as:

$$(\Delta\epsilon_{vd})_{\frac{1}{2}ciclo} = C_1\exp\left(-C_2\left(\frac{\epsilon_{vd}}{\gamma}\right)\right) \tag{5}$$

where $\Delta\varepsilon vd$ is the decrease in volume per half cycle of deformation; $\gamma$ is the shear strain, and $C1$ and $C2$ are constants that can be calculated as:

$$C_1 = \frac{1}{2}C_1^c; \ C_2 = \frac{0.4}{C_1^c}; \ C_1^c = 87\,(N_1)_{60}^{-1.25} \tag{6}$$

These parameters, presented in Table 7, were calculated using the average values of $(N_1)_{60}$, for the first 10 m of soil and the embankment.

**Table 7 Coefficients used for the Finn-Byrne model**

| Element | $C_1^c$ | $C_1$ | $C_2$ |
|---|---|---|---|
| Soil (0 a 10 m) | 0.14 | 0.07 | 2.82 |
| Embankment (0 to 1.3 m) | 0.08 | 0.04 | 4.71 |
| Embankment (1.3 to 2.6 m) | 0.12 | 0.06 | 3.21 |
| Embankment (2.6 to 4.0 m) | 0.19 | 0.10 | 2.07 |

Five scenarios were considered to assess the seismic performance of the road embankment, considering the variations in sea level registered in the study area, as well as the case of high tide, and soil saturation due to rain, Table 8 and Figure 19.

**Table 8 Sea level for each evaluated scenario.**

| Scenario | Tide | Sea level |
|---|---|---|
| 1 | Low tide | 0.00 m |
| 2 | Average sea level | 0.38 m |
| 3 | High tide | 0.68 m |
| 4 | Maximum tide | 1.28 m |

| 5 | Maximum tide and storm | 1.28 m |
|---|---|---|

Seven control points were established in the model to obtain the soil behaviour parameters. Control points were located along the fault surface, at the toe of the embankment, at the highest point of the embankment, and within the soil deposit, Figure 19. Using the concept of static safety factor, $SF_s$, prior the application of the seismic loading, the general stability of the embankment under static conditions was evaluated for each scenario. $SF_s$ was above 2 in all the scenarios considered. The static safety factor was computed based on the concept of the strength over demand concept.

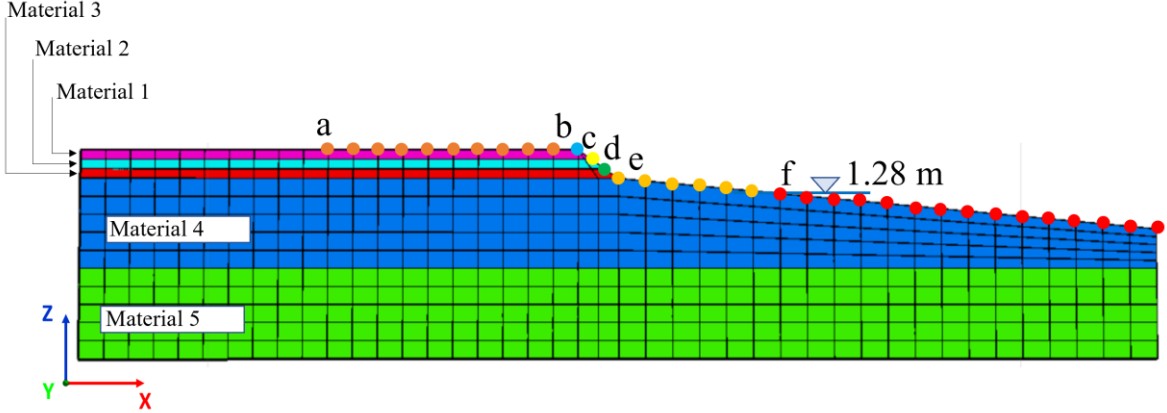

**Figure 19: Model monitoring points.**

Next, an analysis of the seismic response was performed applying the N90E acceleration history, Figure 11. It was observed that the largest vertical displacement, = 73 cm, was recorded in scenario 5, on the crest of the embankment. Figure 20 presents the results of vertical displacement for scenarios 4 and 5 at the end of the earthquake. Figure 21 gives the results of the horizontal displacement for the same cases. The largest horizontal displacement was observed in scenario 5, = 51 cm. Important vertical displacements of the soil were observed, as well as a tendency of lateral displacement of the body of the slope, increasing with depth.

At the control points, Figure 22 shows the pore pressure ratio, $r_u$, obtained for each scenario. A rise in pore pressure was observed at points G and F, in all scenarios. However, with the increase of the sea level, the higher pore pressure ratio at point D, located at the toe of the slope, continues to increase in scenarios 3 and 4, and reaches values above 0.7 in scenario 5, where an additional increase in this ratio was observed at point E, located on the failure surface of the slope.

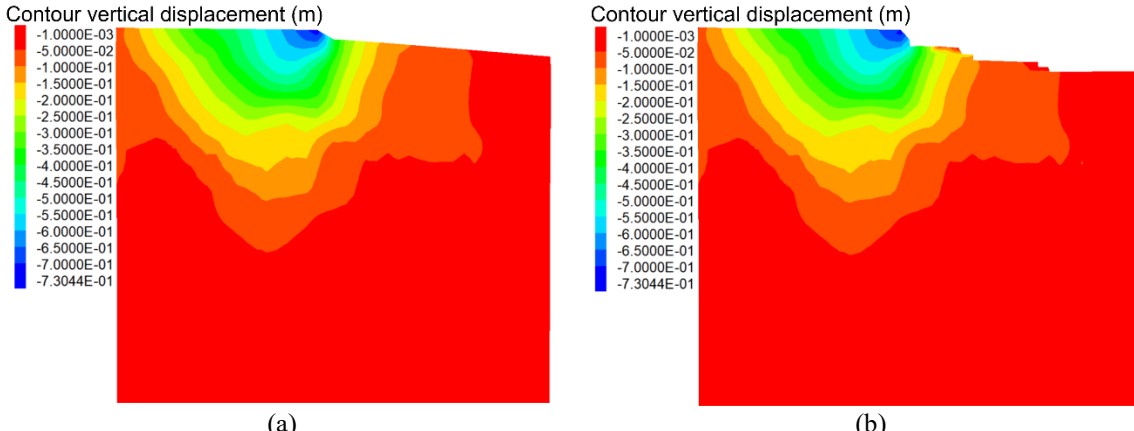

(a)                                    (b)

Figure 20: Permanent vertical displacements for scenarios (a) 4 and (b) 5.

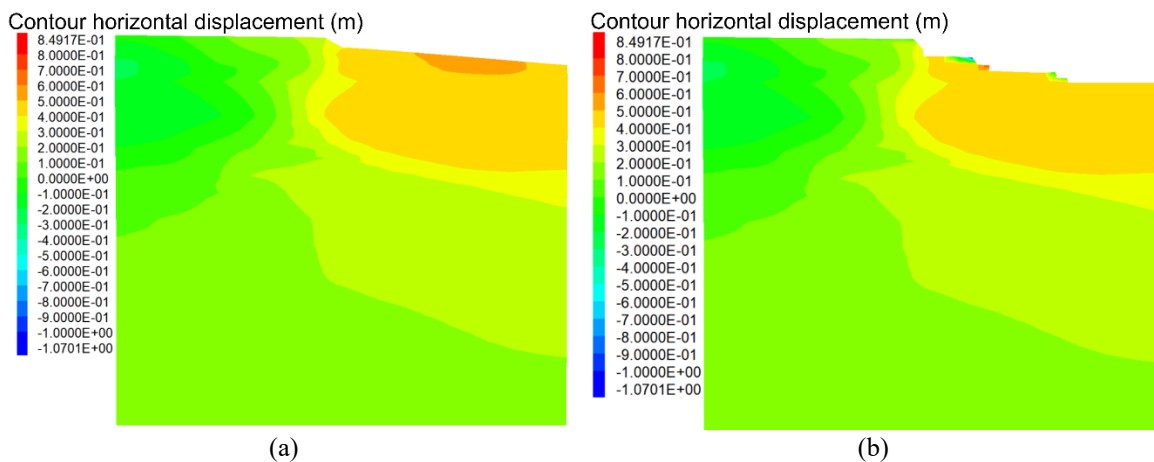

(a)                                    (b)

Figure 21: Permanent horizontal displacements for scenarios (a) 4 and (b) 5.

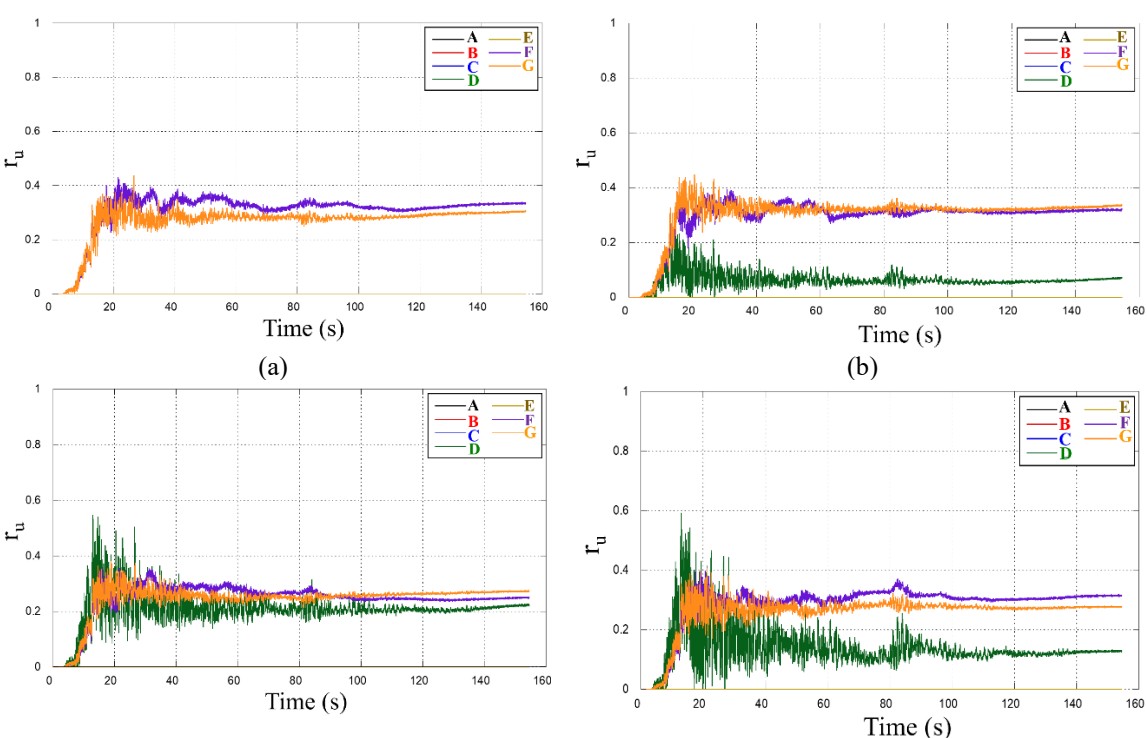

(a)                                    (b)

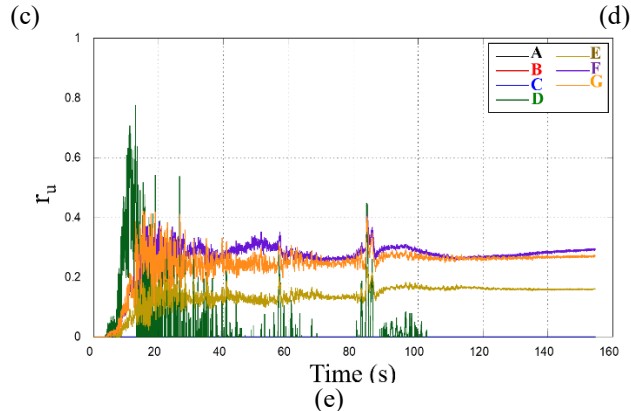

(c)           (d)

(e)

**Figure 22: Variations of pore pressure ratios in scenarios (a) 1, (b) 2, (c) 3, (d) 4 and (e) 5.**

### 3.4. Step 2: simulation of tsunami and wave propagation

The tsunami simulation was carried out using the model implemented in the GeoClaw code (Berger et al., 2011), which is based on solving the non-linear shallow water equations through the numerical method of finite volumes, using adaptive mesh refinement to model small-scale features of the bathymetry as well as structures and coastal elements on a scale of meters (LeVeque, 2011). The shallow water equations are the standard model used for transoceanic tsunami propagation as well as for local inundation: e.g., (Yeh et al., 1994; Titov and Synolakis, 1995; Titov and Synolakis ,1998). In one space dimension these are:

$$h_t + (hu)_x = 0 \tag{7}$$

$$(hu)_t + \left(hu^2 + \tfrac{1}{2}gh^2\right)_x = -ghB_x \tag{8}$$

where $g$ is the gravitational constant, $h(x, t)$ is the fluid depth, $u(x, t)$ is the vertically averaged horizontal fluid velocity. The function $B(x)$ is the bottom surface elevation relative to mean sea level. Where $B < 0$ this corresponds to submarine bathymetry, and where $B > 0$ to topography. GeoClaw code implementation allows the bathymetry and topography to be time-dependent by solving the two-dimensional shallow water equations (LeVeque, 2011):

$$h_t + (hu)_x + (hv)_y = 0 \tag{9}$$

$$(hu)_t + (hu^2 + \tfrac{1}{2}gh^2)_x + (huv)_y = -ghB_x \tag{10}$$

$$(hv)_t + (huv)_x + (hv^2 + \tfrac{1}{2}gh^2)_y = -ghB_y \tag{11}$$

where $u(x, y, t)$ and $v(x, y, t)$ are the depth-averaged velocities in the two horizontal directions, $B(x, y, t)$ is the topography.

The bathymetric and topographic information used was obtained from the GEBCO database, with a resolution of 15 arc seconds. A mesh of 129,600 cells was used, applying 3 levels for mesh refinement, with the finest grids used near the embankment segment, where the grid resolution was 210 m. Considering the characteristics of the fault mechanism of the 1995 Manzanillo earthquake, Table 9, the Okada (1995) fault model was used to estimate the vertical displacement on the seabed caused by the seismic event.

**Table 9 Fault characteristics of the 1995 Manzanillo earthquake**

| Event | Strike | Length (km) | Width (km) | Depth | (km) | Slip | Rake | Dip | Longitude (° | Latitude (°) |
|---|---|---|---|---|---|---|---|---|---|---|
| 09/10/1995 | 309 | 200 e$^3$ | 107 e$^3$ | 10 e$^3$ | | 1.35 | 104 | 14 | -104 | 19 |

Based on the calculated deformations and the characteristics of the earthquake, a tsunami-wave propagation model was run for a simulation period of 1 hour, beginning 15 minutes after the start of the earthquake. Figure 23 shows the simulation results for 1 min. A one-hour simulation period was found for the case study analysed according to records of the duration of the event regarding the wave arriving times (García et al. 1997; Borrero et al. 1997). However, for other cases, longer simulation times could be considered, such as those recommended by ASCE (Robertson, 2017). The authors acknowledge that the grid resolution of the propagation model is a possible research topic for the future. However, a higher resolution was not possible at the time the model was developed. The improvement in the grid spacing would help to reduce uncertainties in the expected flood elevations.

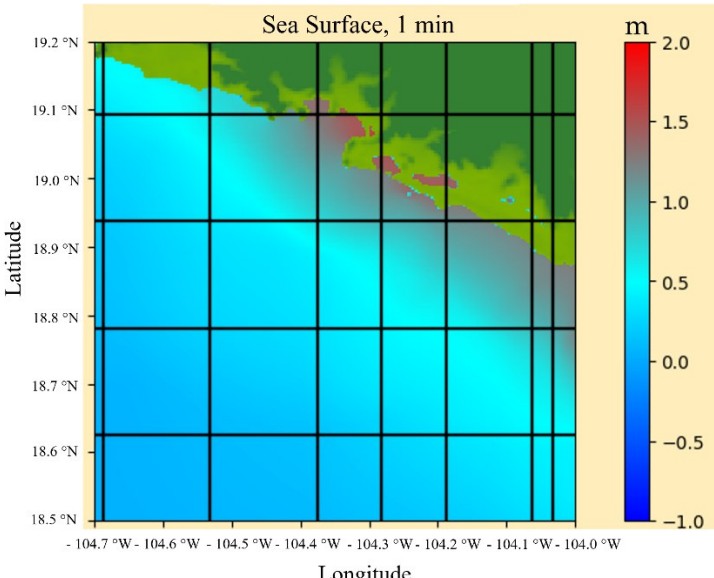

**Figure 23: Sea surface elevation obtained after a simulation time of 1 minute.**

Wave speeds and elevations were obtained at 5 virtual gauges in Santiago Bay, near the road embankment studied. These are shown in Table 10 and Figure 24. Figure 25 shows the change in bathymetry at the gauge locations, as well as the free sea surface elevation, respectively. Figure 26 shows the free sea surface elevation, and $u$ and $v$ the wave velocity components at the gauge nearest to the coast, as well as the wave speed $s$.

**Table 10 Location of virtual gauges in the simulation**

| Gauge | Longitude (°W) | Latitude (°N) |
|---|---|---|
| Gauge 1 | -104.38 | 19.07 |
| Gauge 2 | -104.38 | 19.08 |
| Gauge 3 | -104.38 | 19.09 |
| Gauge 4 | -104.38 | 19.10 |
| Gauge 5 | -104.38 | 19.11 |

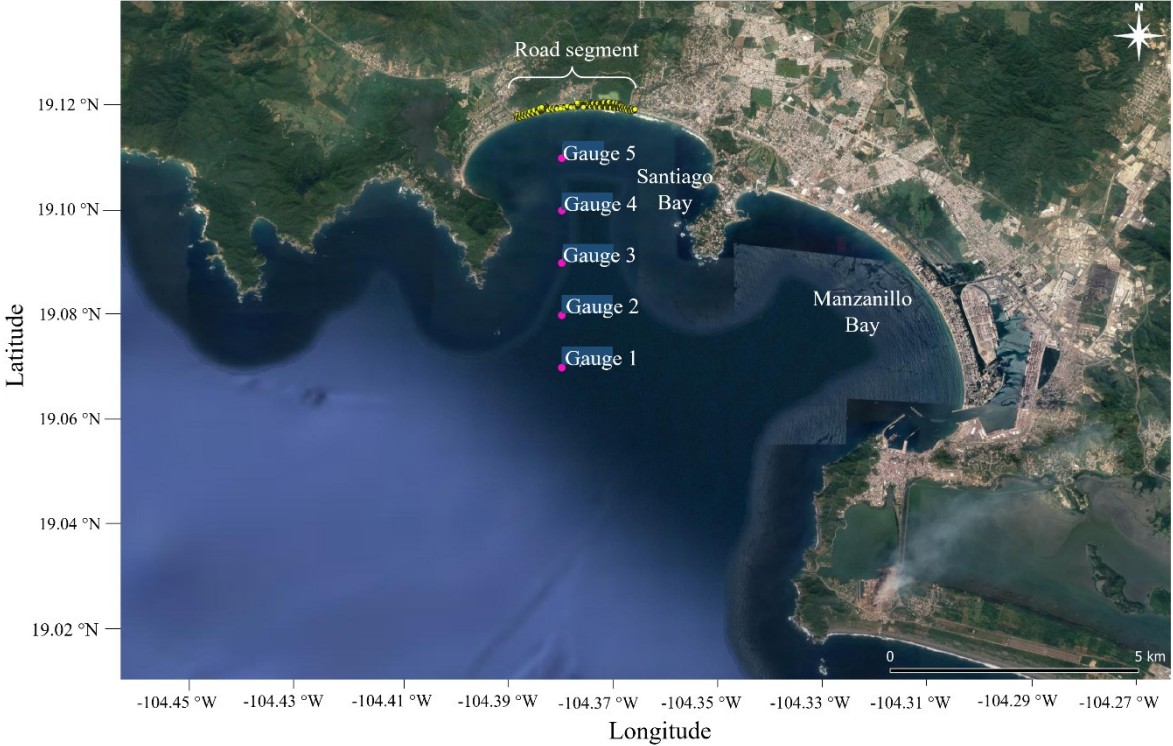

**Figure 24: Location of gauges used to obtain the free sea surface elevations and speeds in the simulation. (© Google Earth).**

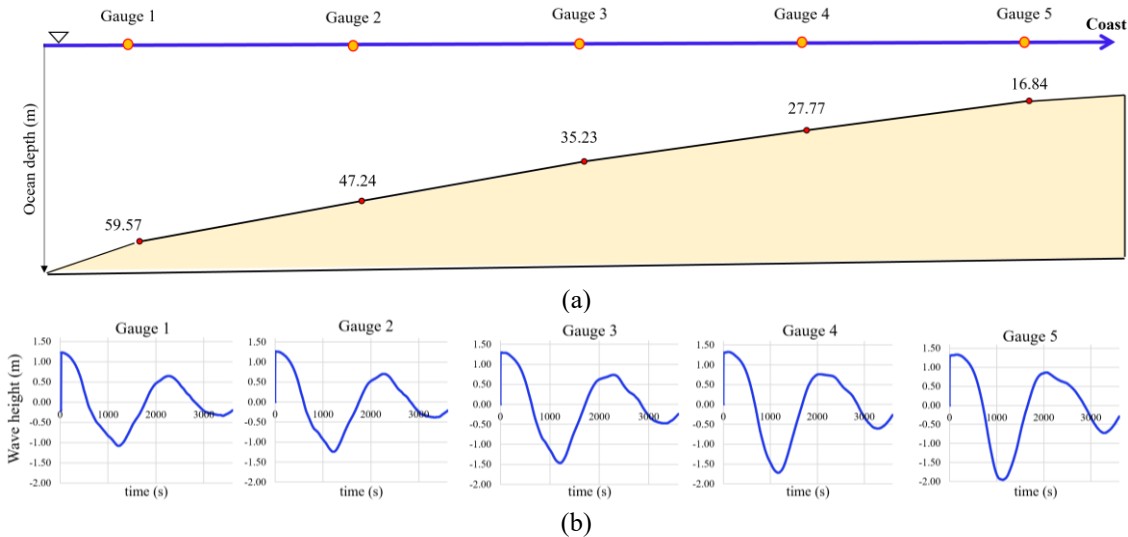

**Figure 25: (a) Ocean depths at the gauge locations (top) and (b) wave height distribution obtained during the**
 **simulation (bottom).**

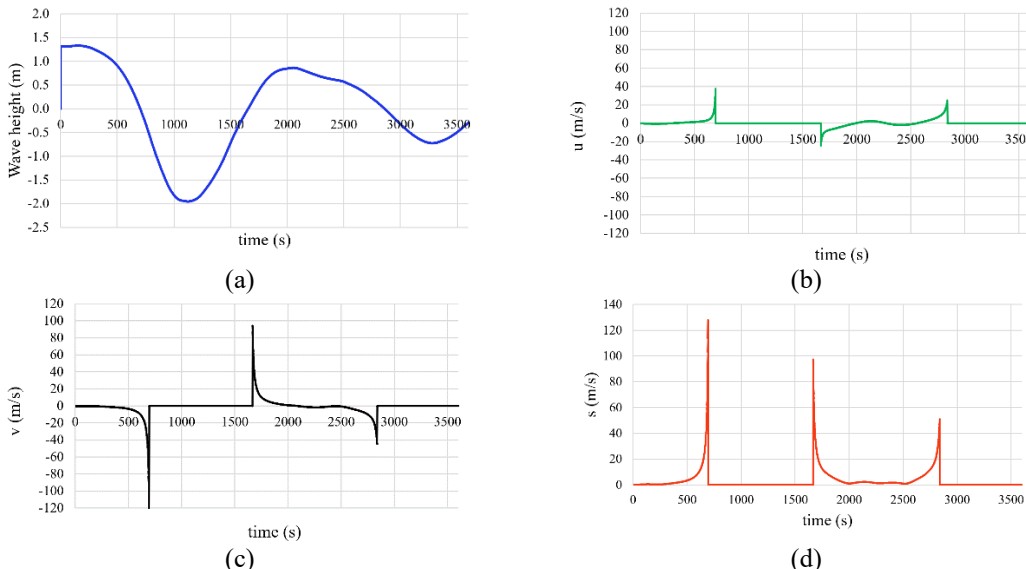

Note: *u*: horizontal velocity, *v*: vertical velocity, *s*: wave speed.

**Figure 26: For the gauge nearest to the coast (Gauge 5): (a) free sea surface elevation (b) velocity *u*, (c) velocity *v*, and (d) wave speed *s*.**

It is worth mentioning that the maximum flood levels obtained coincide with the visual data reported by Avila et al., (2005).

### 3.5.     Step 3: earthquake-tsunami response

Applying a smooth particle hydrodynamics approach, wave-induced hydrodynamic pressures associated with tsunami inundation at the study site were determined over the cross section of Figure 5. The hydrodynamic conditions of the simulation reproduce the flow velocities and elevation of the free sea surface of indicator 5, the closest to the coast (Figure 26). The SPH model used is DualSPHysics, which has been widely used for hydrodynamic forces and to model complex fluid flows (González-Cao et al., 2019; Ye et al., 2019).

**Validation**

To validate the numerical approach, the experimental and numerical model of St Germain (2012) was reproduced, comparing water elevations of the incident wave in two control gauges. A tank 13.17 m long, 2.7 m wide, divided into two sections, and is 1.4 m in height (Figure 27) was considered. The model is made up of a dambreak, consisting of a block of water of 0.85 m height, that is released at time 0 through a swinging gate to travel towards the outflow.

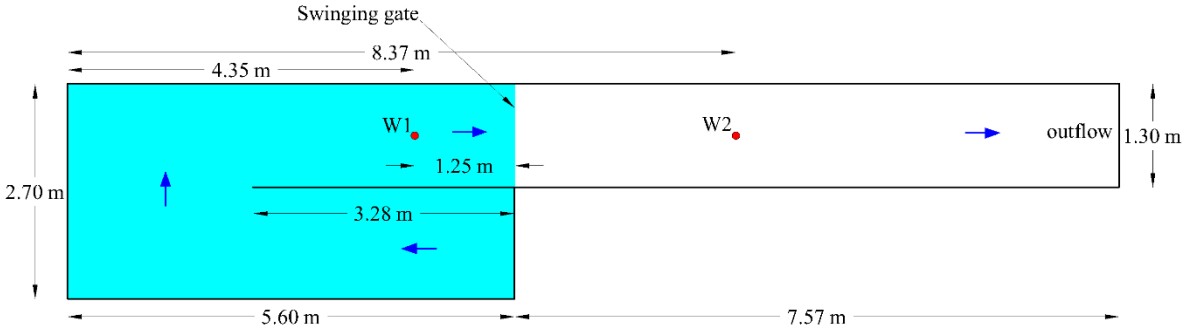

**Figure 27: Experimental model of St Germain et al. (2012).**

The validation SPH simulation parameters are shown in Table 11.

**Table 11 Parameters and main characteristics of the SPH simulation**

| Parameters | Value |
|---|---|
| Kernel Function | Wendland |
| Time-step | Algorithm Verlet |
| Viscosity Artificial | $\alpha=0.01$ |
| Inter-particle spacing (m) | 0.03 |
| Number of particles | 545,929 |
| Simulated time (s) | 10 |
| Time out (s) | 0.1 |
| Computing time (h) | 0.35 |

The simulation results are water heights at points W1 and W2 (Figure 28). The standard deviation of the
DualSPHysics model with respect to the experimental model is 5.4 cm for W1, and 5.2 cm for W2, which are
6.38% and 6.16% with reference to the initial height of the water. The standard deviation of DualSPHysics with
respect to the numerical model of St Germain et al. is 2.9 cm for W1, and 3.5cm for W2, 3.4% and 4.1%
concerning the initial height of the water.

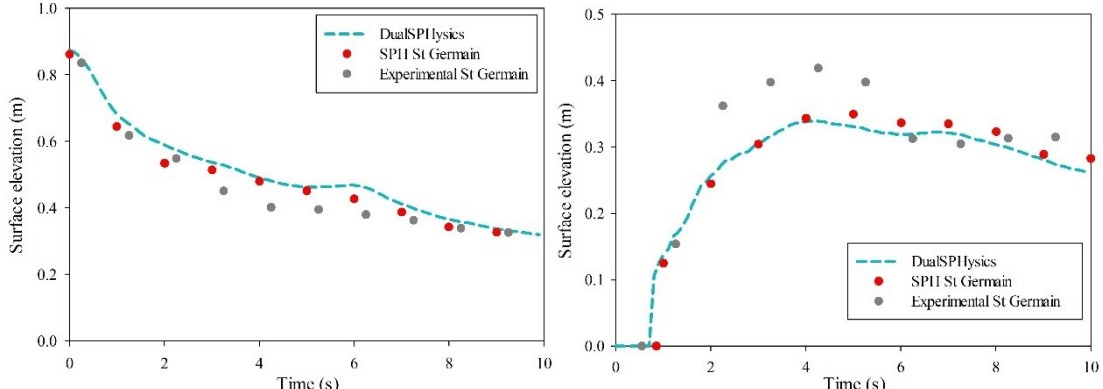

**Figure 28: Water heights at points W1 and W2, St Germain (2012).**

**Analysis of inter-particle spacing sensitivity**

A simulation resolution analysis was performed for the SPH model. The resolution used was the initial size of the lattice nodes of the fluid particles and the fixed particles, defined as inter-particle spacing (dp). The simulation was made through tests with 5 different inter-particle spacings, to obtain the resolution that allows a convergence in the results with a lower computational cost (run time).

The hydrodynamic pressures at the position x = 10 m z = 2.1 m of the different resolutions (dp = 0.1 m, 0.2 m, 0.3 m, 0.4 m, 0.5 m, 0.6 m and 0.7 m) were compared. The agreement of the results obtained with different resolutions is quantified taking as reference the data series of the finest resolution of dp = 0.1 m, by means of the normalized standard deviation ($\sigma_n$), the centered root-mean-square difference ($RMSD$),  and the correlation ($R$), as shown in the Taylor diagram of Figure 29 (as applied in González-Cao et al. 2019 and Klapp et al. 2020).

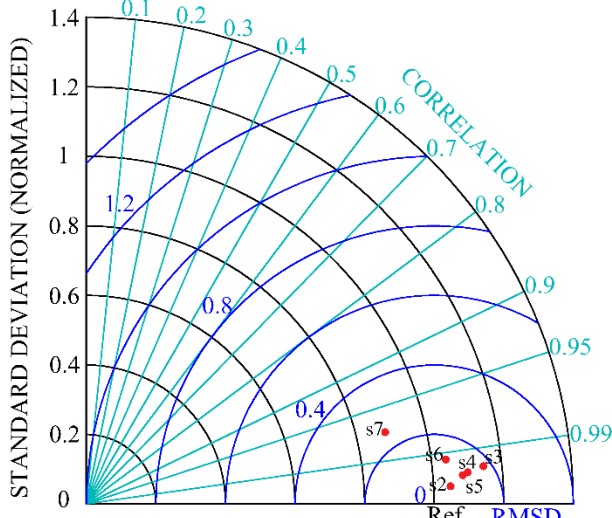

**Figure 29: Taylor diagram of hydrodynamic pressures at x = 10 m and z = 2.1 m, taking as reference the data series of the finest resolution of dp = 0.1 m.**

Details of resolution, run time and number of simulated particles are shown in Figure 30. The parameters and main characteristics of the simulations are shown in Table 12. The simulations were run on NVIDIA 1650 GPU.

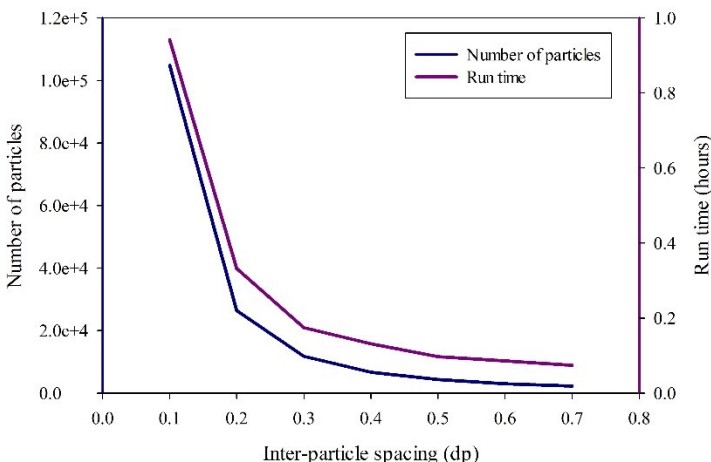


**Figure 30: Number of particles and run time for the different resolutions.**

**Table 12 Parameters and main characteristics of the SPH simulation**

| Parameters | Value |
|---|---|
| Kernel Function | Wendland |
| Time-step | Algorithm Symplectic |
| Viscosity Artificial | $\alpha=0.01$ |
| Inter-particle spacing (m) | 0.1 |
| Number of particles | 104,825 |
| Simulated time (s) | 90 |
| Time out (s) | 0.1 |
| Computing time (h) | 0.94 |

The results of the inter-particle spacing sensitivity analysis show that for all resolutions the correlation is greater than 95% and that improves when the resolution is finer. However, a resolution greater than 0.1 m may not significantly improve the convergence of the results and may increase the computation time. Therefore, a resolution of 0.1 m with a particle number of 104,825 and a computation time of 0.94 h was selected.

Figure 31 shows the pressures acting on the road embankment and Figure 32 shows the pressure evolution at

points a, b, c, d, e and f, obtained in the first 90 s of loading, corresponding to the wave colliding with the slope of the embankment.

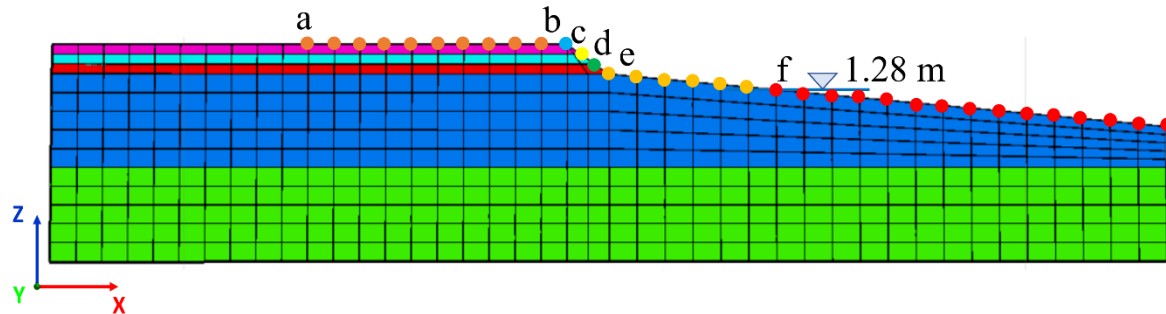

**Figure 31: Initial sea level and control points considered for the earthquake-tsunami loading analysis.**

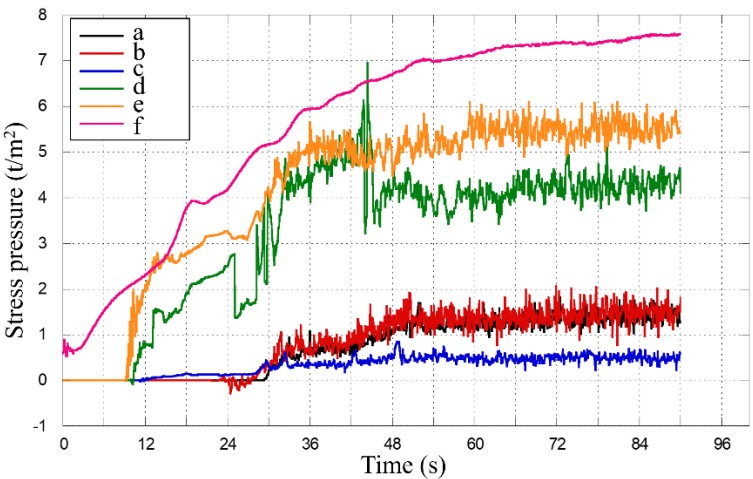


**Figure 32: Loading variation stress as a function of time at points a-f.**

The wave-induced behaviour at the embankment was analysed, applying the same time-step numerical approach of Step 1, and the conditions of Scenario 5, with an initial maximum tide of 1.28 m and saturated soil. From the results obtained in Steps 1 and 2, the corresponding wave elevation of 1.5 m was modelled. In the model, the

hydrodynamic pressures, pore pressures and incremental hydrostatic vertical loading were applied until the steady state in each of the loading points was reached.

Figure 33 shows the hydrostatic and hydrodynamic loading conditions applied at 50 s. The resulting vertical and horizontal displacements, and the safety factor (i.e. capacity over demand) obtained from steady state loading conditions are shown in Figures 34 to 36.

Contour vertical stress (kN/m$^2$)

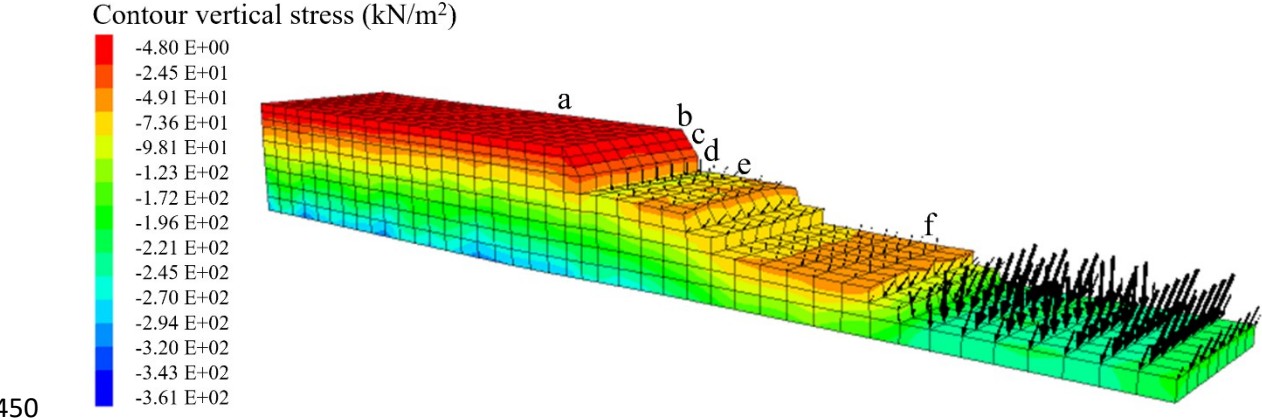


**Figure 33: Loading conditions at points a, b, c, d, e and f, at time 50 s.**

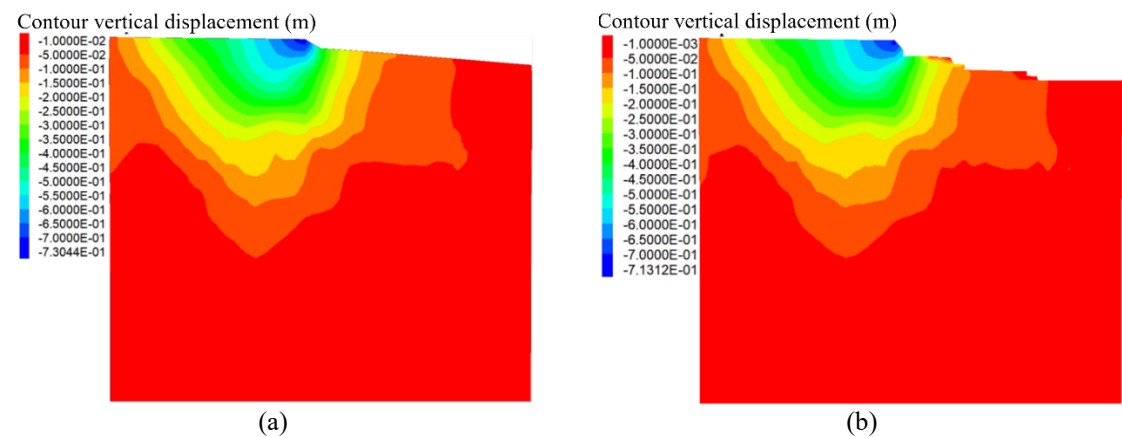

(a)                                                                 (b)

**Figure 34: Vertical displacements contours in meters (a) after seismic loading, (b) after tsunami loading.**

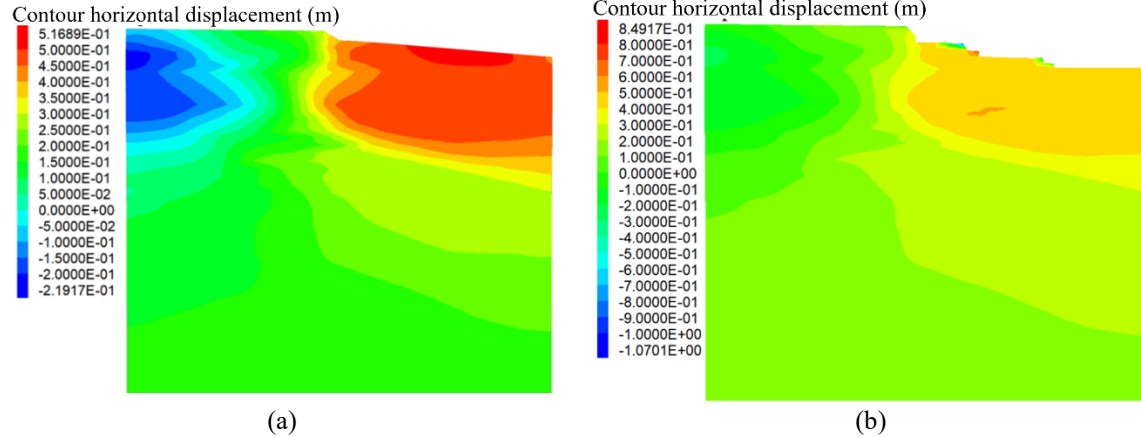

(a)                                                                 (b)

**Figure 35: Horizontal displacement contours in meters (a) after seismic loading, (b) after tsunami loading.**

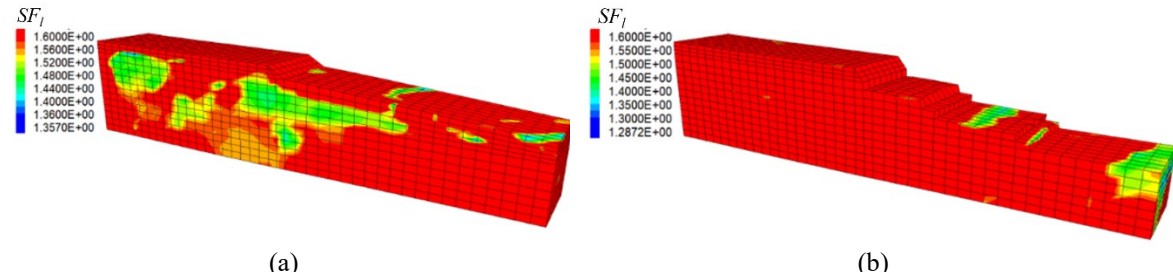

(a)
(b)

**Figure 36: Safety factor versus liquefaction (a) after seismic loading, (b) after tsunami loading.**

The analysis shows that the main effects induced by wave loading are the horizontal displacement of the beach slope, reaching 0.8 m near to point f. No major impacts were seen in the road embankment, where both vertical and horizontal displacements and the liquefaction safety factor show minor variations. However, there was a loss of granular material due to excessive ground deformation in the beach area. This effect was simulated following a time step approach, where zones of the mesh that reach a deformation of 2 m were removed from the model. The hydrodynamic loads were applied to the remaining mesh, which allowed the incremental unloading of the unstable areas in the sand found on the beach slope. The coupled effect of liquefaction and a tsunami could potentially lead to a loss of functionality of the road associated with an approximate 0.7 m displacement of the slope.

## 4.    Conclusions

A sequential methodology was implemented to analyse the response of transportation infrastructure in an earthquake-tsunami. The methodology was applied for Manzanillo, Mexico, where the behaviour of a section of a road embankment was analysed, considering the accumulated effects of the earthquake in terms of horizontal and vertical displacements, and those induced by an increase in sea level after the earthquake. In the case study, different scenarios were analysed depending on the initial elevation of the sea.  The seismic response of the embankment showed that the most critical condition occurred in the scenarios of maximum high tide, and maximum high tide with saturated soil. In these scenarios the vertical displacements were 0.57 m and 0.73 m, respectively, both of which occurred at the embankment crest. The vertical and horizontal displacements can be interpreted as a failure of the embankment, in the sense that they may imply loss of functionality, since these gives a 20% reduction in the height of the structure.

The simulation of wave propagation of a tsunami showed that the level of the free sea surface could increase 1.5 m in the Manzanillo Bay area (Avila-Armella et al., 2005). Further analysis of the earthquake-tsunami behaviour showed that for the scenario of maximum high tide and saturated soil, the highest variation in the horizontal displacements in the ground between the stage at the end of the earthquake and the stage at the end of tsunami-induced flooding, is seen on the beach slope. In the case study, the sequential model shows that seismic loading causes the greatest effects in pore pressure increase and in ground failure due to vertical and horizontal displacements in the embankment.

The sequential approach presented allows soil displacements and strength to be accurately quantified, as well as pore pressure increase derived from an earthquake. The effects also couple with the tsunami arrival, which is not captured in decoupled models. The evaluation of these potential cumulative impacts provides additional information for the design and planning of more sustainable and resilient transportation infrastructure. The method presented is applicable to any coast, as long as there is sufficient information to characterize the site and structures, such as the seismic environment, geotechnics, bathymetry and structural systems. The degree of detail of the information required is of great importance to reduce uncertainty in the results.

From this study it is clear that, as in other parts of the world, the road network in Manzanillo has been built to improve communications, but ignoring many coastal processes, such as the potential presence of a tsunami. Moreover, this infrastructure, was built on coastal dune ridges, inducing coastal squeeze, isolating ecosystems and increasing risk. The planning, design and construction of infrastructure in coastal areas susceptible to earthquakes and tsunamis should be rethought in order to make it compatible with natural processes and thus truly contribute to a better quality of life (Chávez et al., 2021; Silva et al., 2021). In the case of Manzanillo, a pile-driven road is probably a better option.

**Author contribution:**

AR and RS Conceptualization and methodology; AR, NL and AO resources; MV performed the measurements; AR and OA analyzed the data; AR, OA and MV wrote the manuscript draft; RS, EM, NL and AO reviewed and edited the manuscript; SG supervision.

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
