# Peer review of "Modelling the sequential earthquake-tsunami response of coastal road embankment infrastructure"

_Natural Hazards and Earth System Sciences, 2021_

## Author Comment (AC1)

Anonymous Referee #1, 21 Feb 2022

This paper presents a sequential methodology to analyse the response of transportation infrastructure in an earthquake-tsunami event. The content is undoubtedly interesting, and the work developed falls within the scope of Natural Hazards and Earth System Sciences journal. However, given that most of the aspects of the problem have been treated without the due strictness required to a journal paper, the reviewer has serious concerns about the suitability of this paper for publication on NHESS Journal, also in the light of the observations listed below. Therefore, it is suggested to decline the paper.

**Specific Comments:**

**1.- Introduction – this section needs revision, clarification and the inclusion of some recent publications related to lifelines (e.g. the work of J. Williams, 2019, 2020 related to the tsunami vulnerability of critical Infrastructures, etc.) and geotechnics (e.g. the work of Rossetto, T., Goda, K., & De Risi, R., etc...). In this section (at the end of page 2) the models referred to the infrastructure and its interaction with the soil are mixed and needs revision.**

AC: The introduction section was revised, and the following recent references were included:

Williams, J. H., Wilson T.M., Horspool N., Lane E.M., Hughes M.W., Davies T., Le L., Scheele F.: Tsunami impact assessment: development of vulnerability matrix for critical infrastructure and application to Christchurch, New Zealand, Natural Hazards, 96, 1167-1211, https://doi.org/10.21203/rs.3.rs-1104603/v1

Goda, K. (2021). Multi-hazard parametric catastrophe bond trigger design for subduction earthquakes and tsunamis. Earthquake Spectra, 37(3), 1827-1848.

Goda, K., De Risi, R., De Luca, F., Muhammad, A., Yasuda, T., & Mori, N. (2021). Multi-hazard earthquake-tsunami loss estimation of Kuroshio Town, Kochi Prefecture, Japan considering the Nankai-Tonankai megathrust rupture scenarios. International Journal of Disaster Risk Reduction, 54, 102050.

Goda, K., Petrone, C., De Risi, R., & Rossetto, T. (2017). Stochastic coupled simulation of strong motion and tsunami for the 2011 Tohoku, Japan earthquake. Stochastic Environmental Research and Risk Assessment, 31(9), 2337-2355.

**2.- In the different models developed in this study, the software used should be clearly identified and the references added. The models adopted for the soil should be detailed defined.**

AC:  The following paragraphs were included to enhance the model description:

*Site response model*

[revised manuscript text omitted]

**Section 3.4 "Step 2: simulation of tsunami and wave propagation":**

**3.- Page 302-303: GEBCO is not a project of NOAA; this sentence needs revision.**

AC: Line was revised and modified.

"The bathymetric and topographic information used was obtained from the GEBCO database, with a resolution of 15 arc seconds "

**4.- The grid for the simulation considers the displacements presented in previous section. In lines 289-290 it is referred "Important vertical displacements of the soil were observed, as well as a tendency of lateral displacement of the body of the slope, increasing with depth."**

AC: Earthquake induced displacements are considered in the last stage of the modelling framework, the analysis considers the accumulated deformation state of the embankment. This is done by applying the wave-induced hydrodynamic loads to the embankment deformed state. Lines 289-290 refers to earthquake-induced displacements.

**5.- Line 303: a grid with 15 arc seconds (about 300m grid spacing) is not enough to model the flood phase of the tsunami.**

AC: The tsunami simulation was carried out using the model implemented in the GeoClaw code (Berger et al., 2011), which is based on solving the non-linear shallow water equations through the numerical method of finite volumes using adaptative mesh refinement to model small-scale features of the bathymetry as well as structures and coastal elements on a scale of meters. A mesh of 129,600 cells was used, applying 3 levels for mesh refinement, with the finest grids used near the embankment segment, where the grid resolution was 210 m. The authors acknowledge that the grid resolution of the propagation model is a possible research topic for the future. However, a higher resolution was not possible at the time the model was developed. The improvement in the grid spacing would help to reduce uncertainties in the expected flood elevations.

**6.- Line 309: One hour of simulation is not sufficient. The authors should check the recommendations in ASCE 7-16.**

AC: The authors consider that a practical estimation of the combined effect can be obtained from a shorter simulation time for the specific event analysed. A one-hour simulation period was found for the case study analysed according to

records of the duration of the event regarding the wave arriving times (García et al. 1997; Borrero et al. 1997). However, for other cases, longer simulation times could be considered, such as those recommended by ASCE (Robertson, 2017). The authors acknowledge that the grid resolution of the propagation model is a possible research topic for the future. However, a higher resolution was not possible at the time the model was developed. The improvement in the grid spacing would help to reduce uncertainties in the expected flood elevations.

The following paragraph was included:

"Based on the calculated deformations and the characteristics of the earthquake, a tsunami-wave propagation model was run for a simulation period of 1 hour, beginning 15 minutes after the start of the earthquake. Figure 23 shows the simulation results for 1 min. A one-hour simulation period was found for the case study analysed according to records of the duration of the event regarding the wave arriving times (García et al. 1997; Borrero et al. 1997). However, for other cases, longer simulation times could be considered, such as those recommended by ASCE (Robertson, 2017). The authors acknowledge that the grid resolution of the propagation model is a possible research topic for the future. However, a higher resolution was not possible at the time the model was developed. The improvement in the grid spacing would help to reduce uncertainties in the expected flood elevations."

**7.- In this study, if the effect is to be considered as cascade, the variables over time have to account for the cumulative effect in the numerical models.**

AC:  This is correct. This four-stage analysis considers de cumulative effect by modelling both phenomena sequentially. The fourth stage of the analysis considers the accumulated deformation state of the embankment, obtained from the previous stages. The wave-induced hydrodynamic forces are applied to the embankment deformed state. The three analytical approaches are applied dynamically, considering accurately the duration of ground shaking, cyclic loading, pore pressure evolution, wave propagation from the source to the coast and hydrodynamic loading to the embankment. A time-scale diagram was added to clarify the modelling sequence.

[Figure]

**8.- In this work, to evaluate the effect of the tsunami in the infrastructure it is necessary to consider the hydrodynamics variables: height and velocity of propagation of the wave.**

AC: Both hydrodynamics variables: height and velocity of propagation of the wave were considered in the model. The hydrodynamic conditions of the simulation reproduce the flow velocities and elevation of the free sea surface of indicator 5, the closest to the coast. The SPH model used is DualSPHysics, which has been widely used for hydrodynamic forces and to model complex fluid flows (González-Cao et al., 2019; Ye et al., 2019). The validation process was included.

**9.- Figures 24 and 25 need improvement and clarification.**

AC: Figures were modified for clarification.

**Section 3.5 "Step 3: earthquake-tsunami response":**

**10.-The smooth particle hydrodynamics approach (sph) is highly sensitive to the numerical parameters. The smooth particle hydrodynamics approach (sph) is highly sensitive to the numerical parameters. Using this approach without a convergence study, without a validation, is assuming that any output is possible. Moreover, characterize the pressure with sph is still more challenging. Some validation is required in order to obtain reliable results.**

AC: The SPH model validation was added using the experimental and numerical model of St Germain (2012). The following paragraphs were added:

"To validate the numerical approach, the experimental and numerical model of St Germain (2012) was reproduced, comparing water elevations of the incident wave in two control gauges. A tank 13.17 m long, 2.7 m wide, divided into two sections, and is 1.4 m in height (Figure 27) was considered. The model is made up of a dambreak, consisting of a block of water of 0.85 m height, that is released at time 0 through a swinging gate to travel towards the outflow.

The validation SPH simulation parameters are shown in Table 11.

Table 11 Parameters and main characteristics of the SPH simulation

| Parameters | Value |
| --- | --- |
| Kernel Function | Wendland |
| Time-step | Algorithm Verlet |
| Viscosity Artificial | $\alpha=0.01$ |
| Inter-particle spacing (m) | 0.03 |
| Number of particles | 545,929 |
| Simulated time (s) | 10 |
| Time out (s) | 0.1 |
| Computing time (h) | 0.35 |

The simulation results are water heights at points W1 and W2 (Figure 28). The standard deviation of the DualSPHysics model with respect to the experimental model is 5.4 cm for W1, and 5.2 cm for W2, which are 6.38% and 6.16% with reference to the initial height of the water. The standard deviation of DualSPHysics with respect to the numerical model of St Germain et al. is 2.9 cm for W1, and 3.5cm for W2, 3.4% and 4.1% concerning the initial height of the water.

Analysis of inter-particle spacing sensitivity

A simulation resolution analysis was performed for the SPH model. The resolution used was the initial size of the lattice nodes of the fluid particles and the fixed particles, defined as inter-particle spacing (dp). The simulation was made through tests with 5 different inter-particle spacings, to obtain the resolution that allows a convergence in the results with a lower computational cost (run time).

The hydrodynamic pressures at the position x = 10 m z = 2.1 m of the different resolutions (dp = 0.1 m, 0.2 m, 0.3 m, 0.4 m, 0.5 m, 0.6 m and 0.7 m) were compared. The agreement of the results obtained with different resolutions is quantified taking as reference the data series of the finest resolution of dp = 0.1 m, by means of the normalized standard deviation ($\sigma_n$), the centered root-mean-square difference ($RMSD$), and the correlation (R), as shown in the Taylor diagram of Figure 29 (as applied in González-Cao et al. 2019 and Klapp et al. 2020).

Details of resolution, run time and number of simulated particles are shown in Figure 30. The parameters and main characteristics of the simulations are shown in Table 12. The simulations were run on NVIDIA 1650 GPU.

Table 12 Parameters and main characteristics of the SPH simulation

| Parameters | Value |
| --- | --- |
| Kernel Function | Wendland |
| Time-step | Algorithm Symplectic |
| Viscosity Artificial | $\alpha=0.01$ |
| Inter-particle spacing (m) | 0.1 |
| Number of particles | 104,825 |
| Simulated time (s) | 90 |
| Time out (s) | 0.1 |
| Computing time (h) | 0.94 |

The results of the inter-particle spacing sensitivity analysis show that for all resolutions the correlation is greater than 95% and that improves when the resolution is finer. However, a resolution greater than 0.1 m may not significantly improve the convergence of the results and may increase the computation time. Therefore, a resolution of 0.1 m with a particle number of 104,825 and a computation time of 0.94 h was selected."

**11.- How the soil was modelled should be clearly defined.**

AC: The embankment and soil were modelled using the same three-dimensional analysis of stage one, starting from the deformed state model obtained. Wave arrival and interaction with the soil and embankment was analysed applying the resulting vertical and horizontal components of hydrodynamic loads in the corresponding elements of the sand slope and the embankment faces. Loading was applied incrementally, until the steady state of each point was reached, in an overall 1.5 min period.

---

## Author Comment (AC2)

Anonymous Referee #2, 08 Mar 2022

The paper proposes a very relevant and interesting approach due to the concomitant approach of two processes that are usually analysed separately and in isolation: the seismic component and the hydrodynamic component of the tsunami.

The reader will find a new methodological approach to better understand and assess in more detail the possible infrastructural impacts occurring in an area exposed to seismic and tsunami events.

The paper is clear and well ordered, however there are certain aspects in the discussion and presentation of results that need to be improved in order to provide greater clarity in the research. I therefore consider that the paper should be accepted once the following (minor / technical) changes are corrected and the explanation extended.

**1.- A) The title of the paper is too general as it may confuse the reader that the methodology presented is suitable (or may be suitable) for any "urban transport infrastructure", when in fact it applies to a very particular typology called "road embankment" which is a perimeter-oriented exposed on the coast, therefore I recommend changing the title of the paper to "Modelling the sequential earthquake-tsunami response of coastal road embankment infrastructure".**

AC: The title of the paper was modified as "Modelling the sequential earthquake-tsunami response of coastal road embankment infrastructure".

**2.- B) For the tsunami wave simulation part, the authors do not explain in sufficient detail general aspects of the model set-up, the definition of the theoretical vs. real forcing wave, mesh resolution vs. bathymetry, the use of very low resolution bathymetry such as GEBCO, and the transient and 3D processes that the tsunami wave would experience on the coast under study. The authors are therefore invited to make a more detailed discussion of these aspects, especially the implication of approximating the analysis to a single coastal profile in a markedly 3D environment, what considerations/hypotheses are taken into account, is the 2D approximation sufficient, can the bathymetry used adequately represent these detailed processes, is the 2D approximation sufficient, and can the bathymetry used adequately represent these detailed processes? Is the error committed by the use of low resolution bathymetry greater than the quantified results of the simultaneous earthquake+tsunami process? For example.**

AC: The following lines were added for clarification.

"The soil-embankment system was modeled using a three-dimensional approach where the selected coastal profile was considered uniform in a 20 m segment. This was considered the most critical condition of the road because of the embankment height and slope."

"The tsunami simulation was carried out using the model implemented in the GeoClaw code (Berger et al., 2011), which is based on solving the non-linear shallow water equations through the numerical method of finite volumes, using adaptive mesh refinement to model small-scale features of the bathymetry as well as structures and coastal elements on scale of meters (LeVeque 2011). The shallow water equations are the standard model used for transoceanic tsunami propagation as well as for local inundation: e.g., Yeh, Liu, Briggs and Synolakis (1994) and Titov and Synolakis (1995, 1998). In one space dimension these are:

$$h_t + (hu)_x = 0,$$

$$(hu)_t + \left(hu^2 + \frac{1}{2}gh^2\right)_x = -ghB_x,$$

where $g$ is the gravitational constant, $h(x, t)$ is the fluid depth, $u(x, t)$ is the vertically averaged horizontal fluid velocity. The function $B(x)$ is the bottom surface elevation relative to mean sea level. Where $B < 0$ this corresponds to submarine bathymetry and where $B > 0$ to topography. GeoClaw code implementation allows the bathymetry and topography to be time-dependent by solving the two-dimensional shallow water equations:

$$h_t + (hu)_x + (hv)_y = 0,$$

$$(hu)_t + (hu^2 + \frac{1}{2}gh^2)_x + (huv)_y = -ghB_x,$$

$$(hv)_t + (huv)_x + (hv^2 + \frac{1}{2}gh^2)_y = -ghB_y,$$

where $u(x, y, t)$ and $v(x, y, t)$ are the depth-averaged velocities in the two horizontal directions, $B(x, y, t)$ is the topography.

The bathymetric and topographic information used was obtained from the GEBCO database, with a resolution of 15 arc seconds. A mesh of 129,600 cells was used, applying 3 levels for mesh refinement, with the finest grids used near the embankment segment, where the grid resolution was 210 m. Considering the characteristics of the fault mechanism of the 1995 Manzanillo earthquake, Table 9, the (Okada, 1995) fault model was used to estimate the vertical displacement on the seabed caused by the seismic event.

Based on the calculated deformations and the characteristics of the earthquake, a tsunami-wave propagation model was run for a simulation period of 1 hour, beginning 15 minutes after the start of the earthquake. Figure 23 shows the simulation results for 1 min. A one-hour simulation period was found for the case study analysed according to records of the duration of the event regarding the wave arriving times (García et al. 1997; Borrero et al. 1997). However, for other cases, longer simulation times could be considered, such as those recommended by ASCE (Robertson, 2017). The authors acknowledge that the grid resolution of the propagation model is a possible research topic for the future. However, a higher resolution was not possible at the time the model was developed. The improvement in the grid spacing would help to reduce uncertainties in the expected flood elevations."

**3.- C) In the final discussion of the paper, the authors do not really make clear or quantify what improvement is achieved by considering the simultaneous seismic+tsunami methodology, compared to a more traditional, decoupled approach. There is a description that attempts to clarify this point, but it is not entirely clear. Please include an in-deep discussion and a quantification of these effects.**

AC: The following paragraph was added:

"The sequential approach presented allows soil displacements and strength to be accurately quantified, as well as pore pressure increase derived from an earthquake. The effects also couple with the tsunami arrival, which is not captured in decoupled models. The evaluation of these potential cumulative impacts provides additional information for the design and planning of more sustainable and resilient transportation infrastructure."

**4.- D) Finally, the authors do not carry out any self-critical work on the method in relation to the limitations it may have, when trying to apply it to different places in the world, with different coastal protection structures, in markedly 3D environments, etc. If no comment is included in this regard it would seem that the method can only be applied to Manzanillo or areas of the world that are similar in infrastructure (?).**

AC:  The following paragraph was added:

"The method presented is applicable to any coast, as long as there is sufficient information to characterize the site and structures, such as the seismic environment, geotechnics, bathymetry and structural systems. The degree of detail of the information required is of great importance to reduce uncertainty in the results."

Minor changes:
**5.- Figure 3 is not referenced within the text.**

AC: Figure reference was included in the text.

**6.- Figures 6 and 7, could be merged into 1, this would be better understood.**

AC: Figures were merged into one.

**7.- Figure 26. Not clear what the colours are?**

AC: Figure was improved to clarify that colours refer to soil and embankment layers.